# Sparser, Better, Faster, Stronger:
# Sparsity Detection for Efficient Automatic Differentiation

**Adrian Hill**[*]                                                                    *hill@tu-berlin.de*
*BIFOLD – Berlin Institute for the Foundations of Learning and Data, Berlin, Germany*
*Machine Learning Group, Technical University of Berlin, Berlin, Germany*

**Guillaume Dalle**[*]                                                  *guillaume.dalle@enpc.fr*
*LVMT, ENPC, Institut Polytechnique de Paris, Univ Gustave Eiffel, Marne-la-Vallée, France*

**Reviewed on OpenReview:** *https://openreview.net/forum?id=GtXSN52nIW*

## Abstract

From implicit differentiation to probabilistic modeling, Jacobian and Hessian matrices have many potential use cases in Machine Learning (ML), but they are viewed as computationally prohibitive. Fortunately, these matrices often exhibit sparsity, which can be leveraged to speed up the process of Automatic Differentiation (AD). This paper presents advances in *sparsity detection*, previously the performance bottleneck of Automatic Sparse Differentiation (ASD). Our implementation of sparsity detection is based on operator overloading, able to detect both local and global sparsity patterns, and supports flexible index set representations. It is fully automatic and requires no modification of user code, making it compatible with existing ML codebases. Most importantly, it is highly performant, unlocking Jacobians and Hessians at scales where they were considered too expensive to compute. On real-world problems from scientific ML, graph neural networks and optimization, we show significant speed-ups of up to three orders of magnitude. Notably, using our sparsity detection system, ASD outperforms standard AD for one-off computations, without amortization of either sparsity detection or matrix coloring.

## 1 Introduction

### 1.1 Motivation

Machine Learning (ML) has witnessed incredible progress in the last decade, a lot of which was driven by Automatic Differentiation (AD) (Griewank and Walther, 2008; Baydin et al., 2018; Blondel and Roulet, 2024). Thanks to AD, working out gradients by hand is no longer a requirement for training differentiable models. User-friendly software packages like `TensorFlow` (Abadi et al., 2015), `PyTorch` (Paszke et al., 2019) and `JAX` (Bradbury et al., 2018) allow practitioners to quickly experiment with different models and architectures, resting assured that gradients will be computed efficiently and correctly[1] without human intervention

However, while gradient-based optimization has become ubiquitous within ML, the practical use of Jacobians and Hessians remains scarce. Conventional wisdom tells us that for realistic applications, these matrices are too large to handle, since we cannot afford to store $n^2$ coefficients in memory when the number of parameters $n$ reaches millions. A common workaround is to manipulate matrices in form of so-called *lazy* linear operators (Blondel and Roulet, 2024), which are defined only by their action on vectors.

---

[*]Both authors have contributed equally to this work.
[1]While this generally holds true, AD has several pitfalls. Refer to Hückelheim et al. (2024) for a taxonomy.

(a) AD code transformation

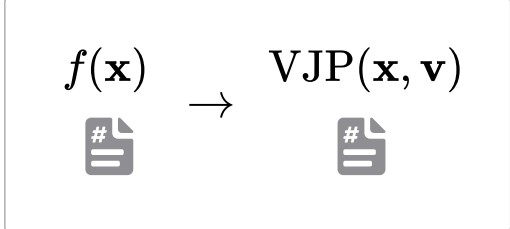

(b) Standard AD Jacobian computation

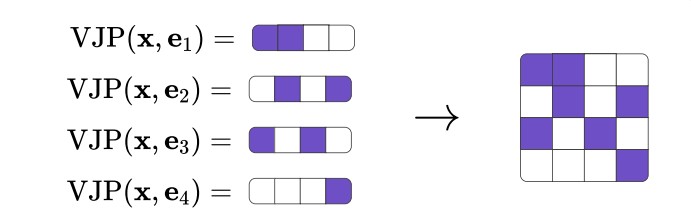

(c) ASD Jacobian computation

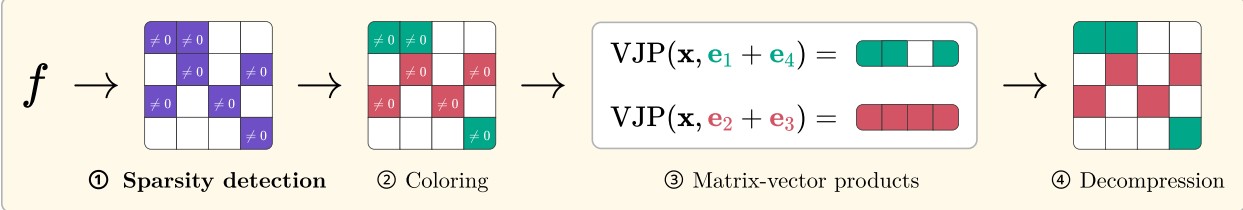

① **Sparsity detection**   ② Coloring   ③ Matrix-vector products   ④ Decompression

Figure 1: Comparison of reverse-mode AD and ASD

(a) Given a function $f$, reverse-mode AD return a function $(\boldsymbol{x}, \boldsymbol{v}) \mapsto \boldsymbol{v}^\top \partial f(\boldsymbol{x})$ computing vector-Jacobian products (VJPs). (b) AD computes Jacobians row-by-row by evaluating VJPs with all basis vectors. (c) ASD reduces the number of VJP evaluations by first detecting a sparsity pattern of non-zero values, then coloring orthogonal rows in the pattern, and simultaneously evaluating VJPs of orthogonal rows. The first step of sparsity detection is the performance bottleneck of ASD and the focal point of this paper. The concepts shown in this figure directly translate to forward-mode, which computes Jacobians column-by-column instead of row-by-row.

Luckily, in numerous applications within ML, most notably in the sciences, Jacobians and Hessians exhibit sparsity — a characteristic that has remained largely ignored by current ML software. By *automatically detecting and leveraging the sparsity* of Jacobians and Hessians (see Figure 1), their automatic computation can be sped up by orders of magnitude in high-dimensional settings. Furthermore, when we materialize these matrices instead of representing them as lazy linear operators, computational advantages such as factorizations and direct linear solves are suddenly unlocked.

## 1.2 Applications

We enumerate concrete scenarios where Jacobians or Hessians appear naturally.

**Newton's method** (Nocedal and Wright, 2006, Chapter 3) is a fast root-finding and optimization algorithm, which is easier to implement with AD. To find a zero of the vector-to-vector function $f \colon \mathbb{R}^n \to \mathbb{R}^m$, Newton's method performs the following iteration:

$$\boldsymbol{x}(t+1) = \boldsymbol{x}(t) - \partial f\left(\boldsymbol{x}\left(t\right)\right)^{-1} f\left(\boldsymbol{x}\left(t\right)\right).$$

To minimize the vector-to-scalar function $f \colon \mathbb{R}^n \to \mathbb{R}$ without constraints, which amounts to finding a zero of the gradient $\nabla f(\boldsymbol{x})$, Newton's method turns into:

$$\boldsymbol{x}(t+1) = \boldsymbol{x}(t) - \nabla^2 f\left(\boldsymbol{x}\left(t\right)\right)^{-1} \nabla f\left(\boldsymbol{x}\left(t\right)\right).$$

In both cases, we need to solve a linear system of equations, involving either a Jacobian matrix $\partial f(\boldsymbol{x})$ or a Hessian matrix $\nabla^2 f(\boldsymbol{x})$ (which is the Jacobian of the gradient). Specifically for optimization, a lot of research effort went into quasi-Newton methods and their limited-memory variants (Nocedal and Wright, 2006, Chapters 6 and 7), which leverage cheap approximations of the inverse Hessian. Still, evaluating the exact Hessian can prove beneficial, for instance to study its spectral properties. In deep learning, the maximum eigenvalue of the training loss Hessian provides insights into the dynamics and stability of gradient descent (Cohen et al., 2020).

**Implicit differentiation** has become more prevalent in ML with the rise of implicit layers (Kolter et al., 2020). When a vector-to-vector function $f(\boldsymbol{x})$ is defined implicitly by conditions of the form $g(f(\boldsymbol{\theta}), \boldsymbol{\theta}) = 0$, the implicit function theorem lets us recover the Jacobian of $f$ by solving yet another linear system, this time with partial Jacobians: $\partial f(\boldsymbol{\theta}) = -\partial_1 g(f(\boldsymbol{\theta}), \boldsymbol{\theta})^{-1} \partial_2 g(f(\boldsymbol{\theta}), \boldsymbol{\theta})$. For unconstrained optimization $f(\boldsymbol{\theta}) = \arg\min_{\boldsymbol{y}} c(\boldsymbol{y}, \boldsymbol{\theta})$, the optimality criterion is $g(f(\boldsymbol{\theta}), \boldsymbol{\theta}) = \nabla_1 c(f(\boldsymbol{\theta}), \boldsymbol{\theta}) = 0$, and so we obtain a partial Hessian to invert. The recent survey by Blondel et al. (2022) gives more insights and examples on implicit differentiation and its connection to AD.

Further applications to probabilistic modeling are discussed in Appendix A. In all of the scenarios mentioned above, we observe that (1) Jacobians and Hessians are useful objects, (2) they are often large and computed with AD, and (3) exact computation is deemed intractable, which seemingly justifies approximations or lazy operators. When the matrices in question exhibit sparsity, we claim that the last item should be examined more closely, and possibly refuted.

### 1.3 Contributions

Despite it being a well-researched area (Griewank and Walther, 2008), sparse differentiation is not widely used in the ML community. We identify two likely reasons for this situation. First, the lack of tooling for *automatic sparsity detection*, which forces potential users to work out the sparsity pattern by hand (a tedious and error-prone process). Second and most importantly, the *separation between the AD and ML scientific communities* (in terms of research groups, publication venues and programming languages), which means that AD advances do not percolate easily into ML circles. For instance, many of the existing sparse differentiation libraries are written in compiled languages like C or Fortran. While powerful, these languages lack the flexibility and iteration speed required by modern ML workflows, which favor dynamic alternatives like Python, R and Julia.

Our contributions are meant to fill these gaps. On the theoretical side, we present operator overloading for sparsity detection in a new light, *reformulating existing techniques* from the AD literature as a binarization of the chain rule. The algorithms we describe have been known for at least two decades, but we hope our new presentation will be more natural for people unfamiliar with the AD literature. Our perspective abstracts away implementation details like computational graphs and the data structures used for bookkeeping. On the practical side, we introduce `SparseConnectivityTracer.jl` (SCT), a highly performant sparsity detection code written in the open-source Julia programming language (Bezanson et al., 2017). SCT allows users to detect both *local* and *global* sparsity patterns, naturally handles dead ends which can occur in traditional graph-based approaches, allows for a flexible selection of data structures for sparsity pattern representations, and enables novel tensor-level overloads that can drastically reduce the amount of operations required to compute sparsity patterns. Drawing on Julia's multiple dispatch paradigm allows it to generate highly performant machine code for each overloaded operator.

We also provide in-depth benchmarks of SCT as part of a full *Automatic Sparse Differentiation* (ASD) system in Julia for the computation of Jacobians and Hessians. Thanks to sparsity detection, ASD can automatically leverage complicated Jacobian and Hessian sparsity patterns without any human involvement. Our benchmarks demonstrate that this ASD implementation outperforms AD at scales previously considered impractical. Notably, it is performant enough to enable speed-ups in one-off computations of Jacobians and Hessians. Thus, while amortizing the computational cost of sparsity detection over several Jacobian and Hessian computations can help, it is not always necessary.

The approach we present does not require any rewriting of user functions, making it compatible with mainstream packages in the Julia ecosystem (such as those for deep learning and scientific ML). We hope that it will provide a blueprint for adaptation in other languages and frameworks, especially in Python which currently lacks sparsity detection and ASD tooling[2].

---

[2]Refer to Appendix B for an overview of ASD implementations in other high-level programming languages.

### 1.4 Notations

Scalar quantities are denoted by lowercase letters $x$, vectors by boldface lowercase letters $\boldsymbol{x}$, and matrices by boldface uppercase letters $\boldsymbol{A}$. Given a vector $\boldsymbol{x}$, its coefficients are written $x_i$. Given a matrix $\boldsymbol{A}$, its columns are written $\boldsymbol{A}_{:,j}$, its rows are written $\boldsymbol{A}_{i,:}$ and its coefficients are written $A_{i,j}$. We use the word "tensor" when we want to refer to either vectors or matrices. The centered dot $\cdot$ stands for a product between two scalars, or the product between a scalar and a tensor. Integer ranges are denoted by $\{1, \ldots, n\}$. For a vector-to-scalar function $f \colon \mathbb{R}^n \to \mathbb{R}$, we write $\nabla f(\boldsymbol{x}) \in \mathbb{R}^n$ for its gradient vector and $\nabla^2 f(\boldsymbol{x}) \in \mathbb{R}^{n \times n}$ for its Hessian matrix. For a vector-to-vector function $f \colon \mathbb{R}^n \to \mathbb{R}^m$, we write $\partial f(\boldsymbol{x}) \in \mathbb{R}^{m \times n}$ for its Jacobian matrix. The partial derivative (or partial Jacobian) of a function with respect to its $k$-th argument is denoted by $\partial_k$, while the total derivative with respect to a variable $v$ is denoted by $d_v$. Unless otherwise specified, all functions considered here are sufficiently differentiable at the point of interest. We will also be interested in the sparsity patterns of gradients, Jacobians and Hessians. The "one" function is defined on numbers as $\mathbf{1}[x] = 1$ if $x \neq 0$ and $\mathbf{1}[x] = 0$ otherwise. Since the sparsity pattern of a generic tensor $\boldsymbol{T}$ is just the one function applied to every element, we write it as $\mathbf{1}[\boldsymbol{T}]$. We use $\vee$ to denote the binary OR operation $a \vee b$. Note that multiplication has priority over the OR operation.

### 1.5 Outline

Section 2 gives a summary of AD and ASD techniques as well as the role of sparsity detection in ASD. Section 3 contains our updated formulation of sparsity detection. Section 4 describes the software implementation of our sparsity detection code SCT. Section 5 showcases numerical experiments for Jacobian and Hessian sparsity detection, as well as resulting ASD benchmarks. Section 6 concludes with future research perspectives.

## 2 Background

### 2.1 Automatic differentiation

As highlighted in the survey by Baydin et al. (2018), AD is a method for computing derivatives that is neither numeric (based on finite difference approximations) nor symbolic (based on algebraic manipulations of expressions). Instead, AD works with *non-standard interpretation* of the source code, allowing it to carry derivatives along with primal values. The reference textbook on the subject is the one by Griewank and Walther (2008), while a more recent treatment is given by Blondel and Roulet (2024). Here we briefly recap the complexities of the main AD *modes*: forward, reverse, and forward-over-reverse.

Let us consider a vector-to-vector function $f \colon \mathbb{R}^n \to \mathbb{R}^m$ and an input $\boldsymbol{x} \in \mathbb{R}^n$. We also fix a perturbation along the input $\boldsymbol{u} \in \mathbb{R}^n$ (tangent) and a perturbation along the output $\boldsymbol{v} \in \mathbb{R}^m$ (cotangent). We call $\tau$ the unit time complexity of evaluating $f(\boldsymbol{x})$ (which may scale with the input size $n$ or output size $m$). Forward mode AD can compute $(\boldsymbol{x}, \boldsymbol{u}) \mapsto \partial f(\boldsymbol{x})\boldsymbol{u}$, called the Jacobian-Vector Product (JVP), in time $\mathcal{O}(\tau)$. Symmetrically, reverse mode AD can compute $(\boldsymbol{x}, \boldsymbol{v}) \mapsto \boldsymbol{v}^\top \partial f(\boldsymbol{x})$, called the Vector-Jacobian Product (VJP), in the same order of time $\mathcal{O}(\tau)$. In particular, the case $m = 1$ implies that gradients are cheap to compute in reverse mode, as observed by Baur and Strassen (1983).

Now let us consider a vector-to-scalar function $f \colon \mathbb{R}^n \to \mathbb{R}$, still with input $\boldsymbol{x}$ and unit time complexity $\tau$. The gradient $g = \nabla f(x)$ can be computed by reverse mode AD. Forward mode AD can then be applied to the computation of $g$. Thus, forward-over-reverse mode AD can compute $(\boldsymbol{x}, \boldsymbol{u}) \mapsto \nabla^2 f(\boldsymbol{x})\boldsymbol{u}$, called the Hessian-Vector Product (HVP), in time $\mathcal{O}(\tau)$. This observation was first made by Pearlmutter (1994) and revisited by LeCun et al. (2012) and Dagréou et al. (2024). Note that some AD systems support *batched evaluation*[3], that is, the joint application of JVPs, VJPs or HVPs to a vector (or batch) of seeds $(\boldsymbol{u}_1, \ldots, \boldsymbol{u}_k)$ all at once.

---

[3]This variant is also commonly called *vector mode*. Given that the seeds themselves can also be vectors, and that the word "mode" already refers to forward or reverse, we hope that our choice of terminology will be less confusing.

## 2.2 Lazy products are not always enough

The routines mentioned above compute matrix-vector products $\boldsymbol{Au}$ involving Jacobians or Hessians, for a cost that is a small multiple of the cost of $f$. But often enough, we need more complex quantities like matrix-matrix products $\boldsymbol{AU}$ or solutions of linear systems $\boldsymbol{A}^{-1}\boldsymbol{b}$. In such cases, a solution based purely on matrix-vector products may be suboptimal, and having access to the full matrix $\boldsymbol{A}$ can yield accelerations.

A matrix-matrix product $\boldsymbol{AU}$ can be computed from $n$ lazy matrix-vector products $\boldsymbol{AU}_{:,j}$, possibly batched. But given the materialized matrix $\boldsymbol{A}$, more efficient procedures exist, for instance in implementations of BLAS Level III (Lawson et al., 1979; Blackford et al., 2002). Similarly, a linear system $\boldsymbol{Au} = \boldsymbol{b}$ can be solved using only matrix-vector products if we resort to iterative methods like the Conjugate Gradient (CG) (Hestenes and Stiefel, 1952) or GMRES (Saad and Schultz, 1986). The precision of these methods is tied to the number of iterations, each of which costs around the same as one function call. On the other hand, given access to $\boldsymbol{A}$, we can use a direct factorization-based solver such as those in LAPACK (Anderson et al., 1999).

When the materialized matrix $\boldsymbol{A}$ is encoded in a sparse format, like Compressed Sparse Column (CSC), Compressed Sparse Row (CSR) or COOrdinate (COO), these conclusions still hold. In particular, complex linear algebra operations can be executed even faster thanks to dedicated libraries. A prominent example is the `SuiteSparse` ecosystem[4] (Davis, 2024), which includes sparse matrix factorizations and direct linear solvers. Since many high-dimensional Jacobians and Hessians are naturally sparse, we can leverage this property to speed up computations if we are able to reconstruct them efficiently.

Numerical linear algebra is not only faster when working on materialized matrices (dense or sparse), it can also be more robust. Iterative solvers shine most when the matrix $\boldsymbol{A}$ is well-conditioned, or when a good preconditioner is known (Stewart, 2022). Outside of these conditions, the gain with respect to direct solvers is less obvious. Finally, it is worth noting that some nonlinear optimization libraries only accept Jacobian/Hessian matrices (usually in sparse formats) for their linear system subroutines, and cannot work with lazy matrix-vector products. Examples include the popular `Ipopt` (Wächter and Biegler, 2006) and `Knitro` (Byrd et al., 2006). All of this suggests that it may be worth paying an initial fee to materialize the matrix, which we recoup as subsequent operations are sped up. This is supported by our experiments in Appendix G and subsection 5.2.

## 2.3 Reconstructing (sparse) matrices from products

One can reconstruct a dense matrix $\boldsymbol{A}$ by taking its products $\boldsymbol{A}_{:,j} = \boldsymbol{Ae}^{(j)}$ (resp. $\boldsymbol{A}_{i,:} = (\boldsymbol{e}^{(i)})^\top \boldsymbol{A}$) with all the basis vectors of the input (resp. output) space. For a function $f : \mathbb{R}^n \to \mathbb{R}^m$, the Jacobian is either built column-by-column with $n$ JVPs, or row-by-row with $m$ VJPs, as shown in Figure 1b. For the Hessian, both options are equivalent due to symmetry. With batched AD, several seeds can be provided at once for the products. In the extreme case where $\boldsymbol{e}^{(1)}, \dots, \boldsymbol{e}^{(n)}$ are all supplied together, batched AD amounts to a matrix-matrix product with the identity $\boldsymbol{AI}_n$. The complexity of this operation scales with the number of basis vectors.

When the matrix $\boldsymbol{A}$ is known to be sparse, this painstaking reconstruction can be greatly accelerated. One option is to perform sparse batched AD (Griewank and Walther, 2008, Chapter 7), essentially computing $\boldsymbol{AI}$ in one pass while dynamically exploiting sparsity inside the function. This approach only applies to a few AD systems because it requires sparsity-aware differentiation of each elementary operation, which is not always implemented. On the other hand, matrix-vector products are always available as the lowest-level primitive of any AD system. To leverage these products generically, we thus focus on compressed evaluation of the matrix (Griewank and Walther, 2008, Chapter 8), which is the standard method for ASD because it can be implemented *on top of any existing AD backend*.

The core idea behind ASD is that, if columns $\boldsymbol{A}_{:,j_1}$ and $\boldsymbol{A}_{:,j_2}$ are structurally orthogonal (they do not share a non-zero coefficient), we can evaluate them together with a single matrix-vector product $\boldsymbol{A}(\boldsymbol{e}^{(j_1)} + \boldsymbol{e}^{(j_2)})$ and then decompress the sum in a unique fashion. An illustration of this procedure is shown in Figure 1c. Finding large sets of structurally independent columns or rows helps lower the number of products necessary

---

[4]https://people.engr.tamu.edu/davis/suitesparse.html

to recover $A$. As shown in the review by Gebremedhin et al. (2005), this matrix partitioning problem is equivalent to a graph coloring problem, where the graph is constructed based on the rows and columns of $A$. Overlapping columns or rows must get different colors, and the goal is to find an assignment which minimizes the total number of colors used. Denoting by $c_n$ the number of colors in the columns and by $c_m$ the number of colors in the rows, we see that only $c_n$ (resp. $c_m$) products are needed to build the matrix column-by-column (resp. row-by-row). For typical sparse matrices, $c_n \ll n$ and $c_m \ll m$, which makes ASD a huge improvement in complexity. For instance, a forward-mode sparse Jacobian can be computed with cost $\mathcal{O}(c_n\tau)$ instead of $\mathcal{O}(n\tau)$.

Crucially, ASD through compressed evaluation requires *a priori knowledge of the sparsity pattern* (where the structural zeros are located). In some special cases, this pattern can be described manually: diagonal and banded matrices are common examples. However, more sophisticated sparsity patterns can emerge from complex code, which makes *sparsity detection* a key component of ASD[5]. If the sparsity pattern does not depend on the input $x$, it can be *reused across several AD calls* at different points. The same goes for the result of coloring. Therefore, runtime measurements usually do not include the cost of this "preparation" step, which is amortized in the long run.

Finally, note that ASD remains accurate *even when the sparsity pattern is overestimated*. If we predict that a coefficient can sometimes be non-zero, but it is in fact always zero, the result will still be correct. Of course, if the coloring involves more colors than strictly necessary, this overestimation makes differentiation slower. Still, this tradeoff might be interesting to save time on sparsity detection.

## 2.4 Related work

The literature on sparse differentiation dates back 50 years. Curtis et al. (1974) first notice that, when computing sparse Jacobians, one can save time by evaluating fewer matrix-vector products. Powell and Toint (1979) extend that insight to sparse Hessians. In the following years, the connection to graph coloring is discovered and several heuristic algorithms are proposed, see Gebremedhin et al. (2005) and references therein. While early works expect the user to provide the sparsity pattern, automated sparsity detection quickly becomes a topic of research. Approaches to sparsity detection can be either dynamic (run-time) or static (compile-time), like for AD itself. In a way, *sparsity detection is equivalent to boolean AD*.

Dixon et al. (1990) propose an operator overloading method based on "doublets" and "triplets" that encode sparse gradients or Hessians. Griewank and Reese (1991) offer an alternative point of view by describing elimination of intermediate vertices in the linearized computational graph. Instead of derivative values, propagating only the sparsity patterns is often more efficient, given that binary information can be encoded into bit vectors (Geitner et al., 1995). Bischof et al. (1996) show that depending on the problem at hand, different sparse storage techniques may be preferred. Griewank and Mitev (2002) suggest a Bayesian criterion to select clever basis combinations and reduce the number of function calls even further. Giering and Kaminski (2006) describe a static transformation of the source code, with rules that echo the aforementioned operator overloading, the bit vector encoding being present as well. Lohoff and Neftci (2024) propose an alternative approach to Jacobian accumulation that is based on reinforcement learning and cross-country elimination (Griewank and Walther, 2008, Chapter 9) and falls outside of forward and reverse mode. Their method leverages sparsity, but lacks support for dynamic control flow, needs to be retrained for each computational graph and is tailored to highly vectorizable functions. Specifically for Hessian sparsity patterns, Walther (2008) extends the operator overloading approach to recognize nonlinear interactions. Walther (2012) discusses a faster variant of their initial algorithm, as well as the choice of the underlying sparse data structures. Meanwhile, Gower and Mello (2012) introduce the edge-pushing algorithm which directly computes a sparse Hessian with its values, bypassing the need for detection, coloring and compressed differentiation. While Walther (2008) requires only forward propagation, Gower and Mello (2012) leverage a reverse pass to increase efficiency: a comparison can be found in Gower and Mello (2014). The edge pushing algorithm is further improved by Wang et al. (2016a); Petra et al. (2018), and shown to be equivalent to the vertex elimination rule of Griewank and Reese (1991) by Wang et al. (2016b).

---

[5]Even when sparsity patterns can be worked out by hand, it helps to compare the result with automated sparsity detection.

In terms of software, most existing sparse differentiation systems are implemented in a low-level language like Fortran or C/C++. Prominent examples include `ADIFOR` (Bischof et al., 1992) and `ADOL-C` (Griewank et al., 1996; Walther, 2009), along with the `ColPack` package for coloring (Gebremedhin et al., 2013). The closed-source MATLAB language also boasts a couple of implementations (Coleman and Verma, 2000; Forth, 2006; Weinstein and Rao, 2017). Furthermore, several algebraic modeling languages for mathematical programming and optimization include some form of sparse differentiation. It is the case at least for `AMPL` (Fourer et al., 1990), `CasADi` (Andersson et al., 2019) and `JuMP.jl` (Dunning et al., 2017), as well as the more recent and GPU-compatible `ExaModels.jl` (Shin et al., 2024).

Nonetheless, a lot of scientific and statistical code is developed directly in open-source, high-level languages like Python, R or Julia. Thus, there is a clear need for fully automatic sparse differentiation libraries which can differentiate user code without the translation layer of a modeling language. In Julia, the current state of the art for sparsity detection relies on a package called `Symbolics.jl` for sparsity detection (Gowda et al., 2019; 2022). As section 5 demonstrates, our contributions inside SCT give rise to a much faster ASD pipeline. A survey of ASD implementation efforts in other high-level programming languages is given in appendix B.

## 3 Detecting sparsity via operator overloading

We now present a revised viewpoint on sparsity detection, exposing the principles behind first-order tracing. In contrast to standard literature (Griewank and Walther, 2008; Walther, 2008), we propose an exposition that does not rely on the notion of computational graph and exploits local sparsity. It also provides a blueprint for easy implementation using a classification of operators, such as the one described in section 4. For second-order tracing, refer to Appendix D.

### 3.1 Principle

Let $f\colon \mathbb{R}^n \to \mathbb{R}^m$ be a vector-to-vector function, and $\boldsymbol{x} \in \mathbb{R}^n$. The Jacobian matrix $\partial f(\boldsymbol{x}) \in \mathbb{R}^{m \times n}$ is obtained by stacking $m$ gradient vectors, since its $i$-th row is the gradient $\nabla f_i(\boldsymbol{x}) \in \mathbb{R}^n$ of the scalar output component $f_i(\boldsymbol{x})$. Thus, we can recover the Jacobian sparsity pattern $\mathbf{1}[\partial f(\boldsymbol{x})] \in \{0,1\}^{m \times n}$ if we know the gradient sparsity pattern $\mathbf{1}[\nabla f_i(\boldsymbol{x})] \in \{0,1\}^n$ of each component.

To achieve this, we use a *tracer*, a number type which contains both a primal value $y(\boldsymbol{x}) \in \mathbb{R}$ (the actual number) and its gradient sparsity pattern $\mathbf{1}[\nabla y(\boldsymbol{x})] \in \{0,1\}^n$ (a binary vector). Importantly, the gradient in question is taken with respect to the input vector $\boldsymbol{x}$. We initialize the procedure by turning every $x_j$ in the input into $(x_j, \boldsymbol{e}_j)$, where $\boldsymbol{e}_j$ is the $j$-th basis vector. Using operator overloading, every intermediate scalar quantity involved in our function $f$ is replaced with a tracer. Thus, at the end of the computation, we recover a tracer $(f_i(\boldsymbol{x}), \mathbf{1}[\nabla f_i(\boldsymbol{x})])$ containing both the primal output and the desired gradient sparsity pattern. All we have left to do is write down the rules on how two such numbers are combined, defining how gradient sparsity patterns propagate.

**Remark 1.** *The tracer type is related to dual numbers, which are a classic ingredient of forward-mode AD. More precisely, it encodes the sparsity pattern of a batched dual number, containing one directional derivative for each input $x_j$. Such batched dual numbers are a way to implement batched forward-mode AD (Revels et al., 2016). One can also see it as the binary version of the sparse doublet in Dixon et al. (1990).*

### 3.2 Propagation rules

Let $\alpha(\boldsymbol{x})$ and $\beta(\boldsymbol{x})$ be two intermediate scalar quantities in the computational graph of the function $f(\boldsymbol{x})$. We compute a new scalar $\gamma(\boldsymbol{x}) = \varphi(\alpha(\boldsymbol{x}), \beta(\boldsymbol{x}))$ by applying a two-argument operator $\varphi$ to $\alpha(\boldsymbol{x})$ and $\beta(\boldsymbol{x})$. Our goal is to express the gradient sparsity pattern $\mathbf{1}[\nabla\gamma(\boldsymbol{x})]$ as a function of the intermediate sparsity patterns $\mathbf{1}[\nabla\alpha(\boldsymbol{x})]$ and $\mathbf{1}[\nabla\beta(\boldsymbol{x})]$. By the chain rule, for any input index $j \in \{1, \ldots, n\}$, the derivative with respect to input $x_j$ can be expressed as follows:

$$\partial_j \gamma(\boldsymbol{x}) = d_{x_j} \varphi(\alpha(\boldsymbol{x}), \beta(\boldsymbol{x})) = \partial_1 \varphi(\alpha(\boldsymbol{x}), \beta(\boldsymbol{x})) \cdot \partial_j \alpha(\boldsymbol{x}) + \partial_2 \varphi(\alpha(\boldsymbol{x}), \beta(\boldsymbol{x})) \cdot \partial_j \beta(\boldsymbol{x})$$

From now on, we omit the dependence on the input to lighten notations, but we keep in mind that everything is evaluated at point $\boldsymbol{x}$:

$$\partial_j \gamma = d_{x_j}\varphi = \partial_1\varphi \cdot \partial_j\alpha + \partial_2\varphi \cdot \partial_j\beta \tag{1}$$

Bringing the indices together shows us that $\nabla\gamma = \partial_1\varphi \cdot \nabla\alpha + \partial_2\varphi \cdot \nabla\beta$. From there, the sparsity pattern emerges, if we extend $\vee$ to represent the elementwise OR.

$$\mathbf{1}[\nabla\gamma] \leq \mathbf{1}[\partial_1\varphi] \cdot \mathbf{1}[\nabla\alpha] \vee \mathbf{1}[\partial_2\varphi] \cdot \mathbf{1}[\nabla\beta] \tag{2}$$

In other words, the propagation of gradient sparsity patterns through the operator $\varphi$ only depends on two binary values: $\mathbf{1}[\partial_1\varphi]$ and $\mathbf{1}[\partial_2\varphi]$. These binary values tell us whether the operator $\varphi$ depends on each of its arguments at the first order.

**Remark 2.** *Note the use of $\leq$ instead of $=$ in Equation 2. It is necessary because $\mathbf{1}[\nabla\alpha]$ and $\mathbf{1}[\nabla\beta]$ forget about actual values, and thus remain blind to accidental cancellations. For instance, this method will always overestimate the sparsity pattern of $\varphi(\alpha, \beta) = \alpha - \beta$ whenever $\alpha(\boldsymbol{x}) = \beta(\boldsymbol{x})$. Fortunately, as discussed above, these overestimates still give rise to correct derivatives inside ASD.*

### 3.3 First-order operator classification

To implement Equation 2, we need to classify every elementary operator $\varphi$ in our programming language depending on whether its partial derivative with respect to each argument is zero. There are two ways to perform this classification: local (accurate) or global (conservative). Local classification takes into account the current value of $\alpha$ and $\beta$, while global classification considers every possible value. In the global case, we effectively replace $\mathbf{1}[\partial_1\varphi(\alpha, \beta)]$ with $\max_{\alpha,\beta}\mathbf{1}[\partial_1\varphi(\alpha, \beta)]$ in Equation 2.

Table 1 gives some examples. The max operator is especially interesting since it comes up in neural networks with activation function $\text{ReLU}(x) = \max(x, 0)$. It is well-known that max (and therefore ReLU) induces local sparsity because it only depends on one of its two arguments: the first one whenever $\alpha \geq \beta$, and the second one whenever $\beta \geq \alpha$. Global sparsity will overlook this subtlety, because there exists a part of the space where $\alpha \geq \beta$ and there exists a part of the space where $\beta \geq \alpha$, so that *both arguments can influence the output* at the first order. Local sparsity allows us to figure out that *only one of them will*. As far as we are aware, local sparsity has rarely been considered in previous works. An example of global and local sparsity patterns on a convolutional layer can be found in Figures 5a and 5b in the appendix.

Global sparsity is still relevant, since it does not require propagating primal values through the computational graph, making it much cheaper to compute. Additionally, it yields a sparsity pattern that is valid over the entire input domain. The cost of the sparsity detection can therefore be amortized over the computation of multiple Jacobians and Hessians at different input points.

| Operator $\varphi(\alpha, \beta)$ | Local $\mathbf{1}[\partial_1\varphi]$ | Local $\mathbf{1}[\partial_2\varphi]$ | Global $\mathbf{1}[\partial_1\varphi]$ | Global $\mathbf{1}[\partial_2\varphi]$ |
|---|---|---|---|---|
| exp, log | 1 | – | 1 | – |
| sin, cos | 1 a.e. | – | 1 | – |
| round, floor, ceil | 0 a.e. | – | 0 | – |
| +, −, *, / | 1 | 1 | 1 | 1 |
| max | $\alpha \geq \beta$ | $\beta \geq \alpha$ | 1 | 1 |
| min | $\alpha \leq \beta$ | $\beta \leq \alpha$ | 1 | 1 |

Table 1: First-order classification of operators
Unary operators have no second argument. "a.e." means "almost everywhere" for the Lebesgue measure

## 4 Software implementation

We implement the tracer number types described in the previous section in an open-source software package called SparseConnectivityTracer.jl (SCT). For global sparsity detection, SCT implements two types

of tracers: gradient tracers, which hold a gradient sparsity pattern $\mathbf{1}[\nabla y(\boldsymbol{x})]$, and hessian tracers, which additionally hold a hessian sparsity pattern $\mathbf{1}[\nabla^2 y(\boldsymbol{x})]$. As outlined in subsection 3.3, local sparsity detection requires an additional wrapper type that holds and propagates the value of the primal computation. The choice of data structures used to represent sparsity patterns $\mathbf{1}[\nabla y(\boldsymbol{x})]$ and $\mathbf{1}[\nabla^2 y(\boldsymbol{x})]$ is flexible and can be altered to fit the problem at hand, as we will discuss in subsection 4.1.

To add an operator overload, a given operator must be classified according to Table 1. SCT then automatically generates performant code that implements Equation 2, as well as correctness tests that check the classification against the forward mode AD system `ForwardDiff.jl` (Revels et al., 2016)[6] . The generated operator overloads work on generic Julia code that supports `Real` numbers. By making use of Julia's *multiple dispatch* paradigm, external software packages automatically call SCT's overloads when they are evaluated with our tracer types, requiring no code modification. As a result, it is already used by the SciML ecosystem[7] (Julia's equivalent of `SciPy`), e.g. for nonlinear root-finding (Pal et al., 2024) and constrained optimization (Dixit and Rackauckas, 2023). Furthermore, we make SCT's code generation utilities available for third party packages, enabling custom operator overloads.

### 4.1  Sparsity pattern representations

For the purpose of mathematical exposition, it was convenient to define gradient and Hessian sparsity patterns as sparse binary tensors $\mathbf{1}[\nabla y(\boldsymbol{x})] \in \{0,1\}^n$ and $\mathbf{1}[\nabla^2 y(\boldsymbol{x})] \in \{0,1\}^{n \times n}$. In SCT, multiple data structures can be flexibly used to represent these sparsity patterns. One approach is to represent sparsity patterns as the sets of (pairs of) indices of non-zero entries: $\mathbf{1}_{\mathtt{set}}[\nabla y(\boldsymbol{x})] := \{i \in \{1, \dots, n\}$ such that $\partial_i y(\boldsymbol{x}) \neq 0\}$ and $\mathbf{1}_{\mathtt{set}}[\nabla^2 y(\boldsymbol{x})] := \{(i, j) \in \{1, \dots, n\}^2$ such that $\partial^2_{ij} f(\boldsymbol{x}) \neq 0\}$. Every operation on sparse binary tensors has an equivalent on index sets. The elementwise `OR` $\mathbf{1}[\nabla\alpha] \vee \mathbf{1}[\nabla\beta]$ is a union $\mathbf{1}_{\mathtt{set}}[\nabla\alpha] \cup \mathbf{1}_{\mathtt{set}}[\nabla\beta]$. Meanwhile, the outer product `OR` $\mathbf{1}[\nabla\alpha] \vee \mathbf{1}[\nabla\beta]^\top$ is a Cartesian product $\mathbf{1}_{\mathtt{set}}[\nabla\alpha] \times \mathbf{1}_{\mathtt{set}}[\nabla\beta]$. We can thus translate Equation 2 as $\mathbf{1}_{\mathtt{set}}[\nabla\gamma] = \mathbf{1}[\partial_1\varphi] \cdot \mathbf{1}_{\mathtt{set}}[\nabla\alpha] \cup \mathbf{1}[\partial_2\varphi] \cdot \mathbf{1}_{\mathtt{set}}[\nabla\beta]$.

To implement it, various data structures for sets, such as hash tables or bit vectors, can be used. In the end, the right choice of set data structure will depend on the performance of unions and iteration, as remarked by Walther (2012). For small problems up to a few hundreds of inputs, bit vectors tend to be the most efficient, as each index requires only one bit of memory. Additionally, unions are fast, amounting to an `OR` operation on bits. The downside of bit vectors is that their memory requirement is constant, regardless of sparsity, and grows with the number of inputs. For sparse computations with very large inputs, hash tables that store indices as integers can therefore be more efficient. Going beyond sets, vectors of indices can be used that allow for duplicated storage. The upside is that unions are just a vector concatenation, which makes them very fast, but a final de-duplication step is always needed. The optimal sparsity pattern representation depends on the dimensions of the problem, the sparsity level, and more generally the structure of the computational graph. Since there is no universal right answer, our generic implementation of tracer types allows users to select the sparsity pattern representation that best suits their task. All sparsity pattern representations, including data structures for set types, can additionally be extended by users through multiple dispatch.

### 4.2  Tensor-level overloads

For the detection of global sparsity patterns, overloads are not exclusively implemented on a scalar level, but also on tensors of tracers. This allows us to bypass the original scalar computational graph for increased computational performance without any loss of precision. For example, when multiplying two matrices of tracers, instead of falling back to elementwise multiplication and addition using scalar overloads, we can make use of the fact that both multiplication and addition have an identical first-order classification (see Table 1). This allows us to reorder operations, first computing elementwise `OR` operations along rows and columns of both matrices respectively. For two matrices of size $(n \times p)$ and $(p \times m)$, this reduces the amount of `OR` operations from $n \cdot m \cdot (2p-1)$ to $(n+m)(p-1) + n \cdot m$, leading to a significant increase in performance. An in-depth derivation of this is given in appendix E.

---

[6]The same holds for second-order tracing using Table 3 for classification, generating code according to Equation 4.

[7]https://sciml.ai/

Similar overloads are implemented for common matrix operators like matrix norms and determinants. For functions that depend non-trivially on all inputs, like the determinant, a conservative sparsity pattern can be computed by returning the union of all input sparsity patterns, thus bypassing expensive operations.

### 4.3 Limitations

While our local tracer types fully support control flow statements, our global tracer types only support a very limited subset[8] due to their lack of a primal value. Common boolean functions (e.g. `iszero`, `isfinite`) also aren't supported. This is by design: if a global tracer were to enter a branch in code, the returned pattern wouldn't be guaranteed to be correct, as a conservative pattern requires the evaluation of all branches. While overestimates of sparsity patterns are acceptable (as described in subsection 2.3), underestimates can lead to an erroneous coloring of orthogonal rows or columns and subsequently wrong ASD computations. Therefore, it is preferable to have global tracers throw an error rather than have them silently return incorrect sparsity patterns. Fortunately, this issue is easily circumvented by leveraging operator overloads. Building on the example from subsection 3.3, the function $\text{ReLU}(x) = \max(x, 0)$ normally requires a comparison of primal values, branching to return either the tracer $x$ or 0. By classifying the first-order derivative of ReLU as *not globally sparse*, SCT is able to generate code for global tracers that circumvents this comparison, directly returning $x$ which yields the most conservative pattern estimate.

The second limitation of SCT is its current lack of GPU support. As described in subsection 4.1, sparsity pattern representations of large inputs currently make use of hash tables or bit vectors, which dynamically allocate memory. While statically sized index sets should in theory be able to run on GPUs, this work is still pending. On top of this issue, GPU parallelized computations in deep learning typically make use of vectorized code, which often results in block-wise sparsity patterns. Such patterns can benefit from a specialized implementation of sparsity detection that does not work on the scalar level. Still, scalar-level implementations such as ours are very relevant for scientific ML, which can involve a lot of fine control flow and individual indexing.

## 5   Numerical experiments

To benchmark SCT's performance in sparsity detection, we compare it to the previous state-of-the-art in Julia[9], which relies on `Symbolics.jl`. When measuring the benefits of SCT in the broader computation of Jacobians and Hessians, we make use of two additional open-source Julia packages inside a single ASD pipeline. The first is `SparseMatrixColorings.jl` (Montoison et al., 2025), which implements and improves coloring algorithms from Gebremedhin et al. (2005; 2007; 2009; 2013). The second is `DifferentiationInterface.jl` (Dalle and Hill, 2025a), a common interface for AD and ASD, which allows users to switch the sparsity detection method between SCT and `Symbolics.jl` in one line of code. A brief introduction to sparse matrix coloring can be found in Appendix C. A demonstration of the API and its usage is given in Appendix F.

In some benchmarks, we distinguish between *prepared* and *unprepared* AD or ASD. Preparation refers to the parts of the pipeline that can be amortized across several computations. For AD it is often negligible, but for ASD preparation includes the costly sparsity detection and coloring, whose results can only be reused if they are input-agnostic. *Prepared* benchmarks are representative for applications which require the computation of multiple Jacobians of the same function, *unprepared* benchmarks correspond to one-off computations or cases where the sparsity pattern could conceivably change between runs.

All experiments and benchmarks were run using Julia 1.11 on an Apple M3 Pro CPU with 36 GB of RAM. The plots are rendered using `Makie.jl` (Danisch and Krumbiegel, 2021). While the benchmarks in the main body of this paper focus on sparse Jacobians, *further experiments* on sparse Jacobians with linear solves, sparse Hessians, and implicit differentiation on Graph Neural Networks can be found in appendices G, H and 5.2 respectively. To ensure reproducibility, we provide the complete

---

[8] For example, Julia's `ifelse` function is supported, since SCT overloads it, but regular if-else blocks aren't.
[9] Since sparsity detection is language-specific, benchmarking across languages is difficult.

source code and matching Julia environments in our public repository `https://github.com/adrhill/sparser-better-faster-stronger/` (Hill, 2025).

## 5.1 Jacobian computation: Brusselator

### 5.1.1 Jacobian sparsity detection

To compare the performance of SCT with `Symbolics.jl`, we benchmark on the same example as Gowda et al. (2019), the *Brusselator* semilinear partial differential equation (PDE), which describes the spatial evolution of an autocatalytic chemical reaction (Prigogine and Lefever, 1968). The PDE is discretized to $N \times N \times 2$ coupled ordinary differential equations using finite differences. Selected sparsity patterns are shown in appendix G.1. The Brusselator benchmark is representative for the application of ASD to the field of scientific machine learning, where it can be used to accelerate the computation of Jacobians in hybrid Neural ODEs (Chen et al., 2018). Instead of training neural networks to learn the full dynamics of a dynamical system from data, Rackauckas et al. (2021) study the incorporation of mechanistic priors, which in turn exhibit sparsity in the resulting Jacobian.

Table 2 measures the wall time of Jacobian sparsity detection depending on the dimensionality $N$ of the discretized Brusselator PDE. SCT outperforms the state of the art by one order of magnitude on large problems and two orders of magnitude on small problems.

| Problem | | | Sparsity | | Sparsity detection[1] | | |
|---|---|---|---|---|---|---|---|
| N | Inputs | Outputs | Zeros | Colors[2] | Symbolics | SCT[3] | |
| 6 | 72 | 72 | 91.67% | 9 | $5.07 \cdot 10^{-3}$ | $\mathbf{2.10 \cdot 10^{-5}}$ | **(241.5)** |
| 12 | 288 | 288 | 97.92% | 10 | $2.12 \cdot 10^{-2}$ | $\mathbf{8.85 \cdot 10^{-5}}$ | **(240.0)** |
| 24 | 1152 | 1152 | 99.48% | 10 | $7.48 \cdot 10^{-2}$ | $\mathbf{3.92 \cdot 10^{-4}}$ | **(190.8)** |
| 48 | 4608 | 4608 | 99.87% | 10 | $3.08 \cdot 10^{-1}$ | $\mathbf{1.96 \cdot 10^{-3}}$ | **(157.2)** |
| 96 | 18432 | 18432 | 99.97% | 10 | $1.45 \cdot 10^{0}$ | $\mathbf{1.71 \cdot 10^{-2}}$ | **(84.5)** |
| 192 | 73728 | 73728 | 99.99% | 10 | $7.19 \cdot 10^{0}$ | $\mathbf{2.44 \cdot 10^{-1}}$ | **(29.5)** |

[1] Wall time in seconds.
[2] Number of colors resulting from greedy column coloring.
[3] In parentheses: Wall time ratio compared to Symbolics.jl's sparsity detection (higher is better).

Table 2: Performance comparison of Jacobian sparsity detection on the Brusselator PDE.

### 5.1.2 Jacobian computation

We now benchmark the full computation of dense and sparse Jacobians on the same Brusselator PDE. For both AD and ASD, the forward-mode backend `ForwardDiff.jl` (Revels et al., 2016) is used to evaluate JVPs. The wall time breakdown of Figure 2 shows that sparsity detection previously was a significant performance bottleneck. For small to medium problems, sparsity detection using `Symbolics.jl` takes up more wall time than the full AD Jacobian computation, which is no longer true for SCT.

The same measurements give rise to the comparison in Figure 3. For large problems, prepared ASD using SCT accelerates the computation of Jacobians by more than three orders of magnitude compared to classical AD. More surprisingly, on all but the smallest Brusselator problem ($N = 6$), one-off unprepared ASD using SCT also outperforms prepared AD. The performance of SCT therefore opens performance gains via one-off ASD to more settings. Detailed benchmark timings are given in appendix G.2.

## 5.2 Implicit differentiation: GNNs

Because it gives access to a materialized Jacobian matrix, ASD unlocks the use of direct linear solvers instead of iterative ones. Among other applications, this can speed up differentiation of implicitly-defined neural

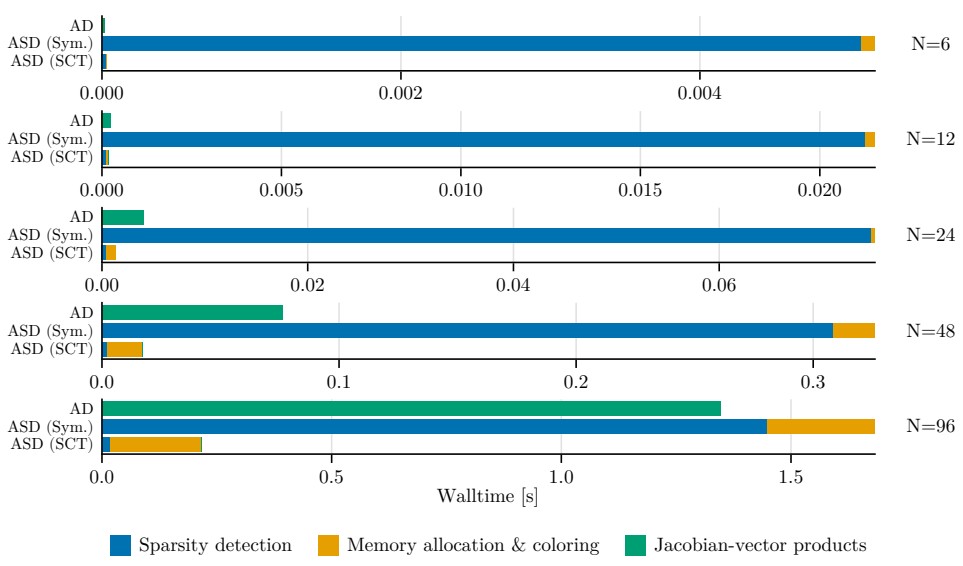

Figure 2: Performance decomposition of unprepared AD and ASD Jacobian computations on small discretizations of the Brusselator PDE. Using `Symbolics.jl` (Sym.), sparsity detection is the performance bottleneck of ASD

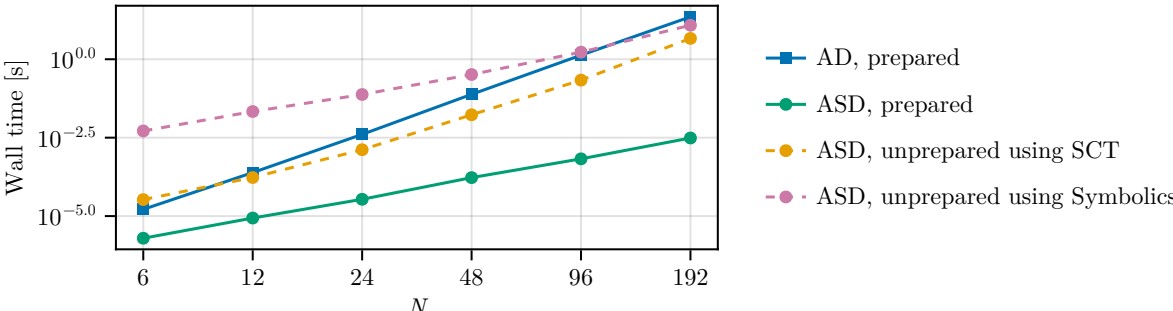

Figure 3: Performance comparison of AD and ASD Jacobian computations on the Brusselator PDE.

network layers. We demonstrate it here with implicit Graph Neural Networks (GNNs), as introduced by Gu et al. (2020).

Consider a graph $\boldsymbol{A}$ represented by its adjacency matrix, and a matrix of vertex features $\boldsymbol{U}$. To compute vertex embeddings and perform predictions, standard GNNs apply the following update a fixed number of times:

$$\boldsymbol{X}^{(t+1)} = \phi(\boldsymbol{W}^{(t)}\boldsymbol{X}^{(t)}\boldsymbol{A} + \boldsymbol{\Omega}^{(t)}\boldsymbol{U}).$$

Here, $\phi$ is a nonlinear activation, while $\boldsymbol{W}^{(t)}$ and $\boldsymbol{\Omega}^{(t)}$ denote layer-specific weights. Conversely, an implicit GNN layer solves the fixed-point equation

$$\boldsymbol{X} = \phi(\boldsymbol{W}\boldsymbol{X}\boldsymbol{A} + \boldsymbol{\Omega}\boldsymbol{U})$$

using an unbounded number of iterations. Instead of backpropagating gradients through these iterations, which is very costly, Gu et al. (2020) suggest to use implicit differentiation for computing $\partial_{\boldsymbol{W}}\boldsymbol{X}$ and $\partial_{\boldsymbol{\Omega}}\boldsymbol{X}$. They derive an efficient formula to solve the linear system inside the implicit function theorem, but in general one could also use generic iterative or direct solvers. For direct solvers, leveraging sparsity is paramount even in moderate dimensions, because most graphs have sparse adjacency structures. We implement and compare both variants using the package `ImplicitDifferentiation.jl` (Dalle, 2022). This library allows the

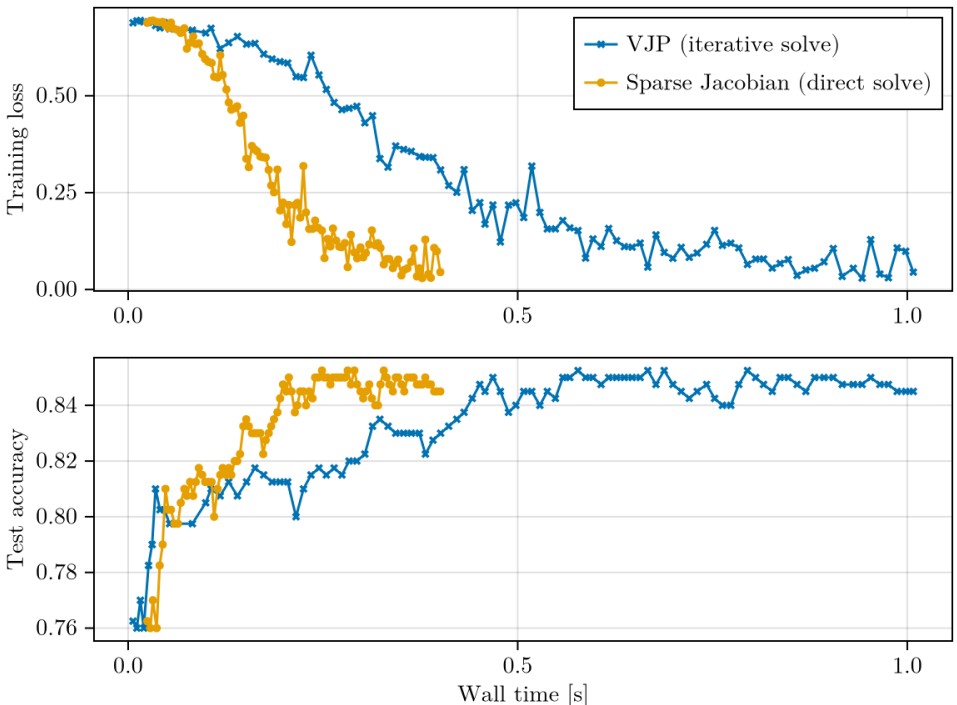

Figure 4: Training an implicit GNN with implicit differentiation, using either iterative solves (with AD) or direct linear solves (with ASD)

use of iterative solvers (based on JVPs or VJPs) as well as direct solvers (based on dense or sparse matrices) in the linear system of equations given by the implicit function theorem. The specific iterative solver used there is `gmres` from `Krylov.jl` (Montoison and Orban, 2023), with its default parameters.

Our test case is a simplified versions of the chains dataset (Gu et al., 2020, Appendix E.1). We perform node classification on a chain of $l = 10$ nodes with 100-dimensional features, where only the first feature of the first node of the chain carries meaningful information (0 or 1 depending on the class to predict). The architecture is an implicit GNN with 16 hidden neurons and ReLU activation, followed by dropout and a linear prediction layer with cross-entropy loss. For the sake of simplicity, some implementation details differ from Gu et al. (2020). The most important ones are listed here, we refer the reader to our experimental code for the rest. We added some small noise to the 99 non-informative features, whereas they were uniformly zero in the original paper. We replaced the orthogonal projection onto the $\ell_1$ ball with a simple scaling of the weights matrix $W$ based on the spectral radius of $A$. This also guarantees that the fixed-point iteration won't diverge. We build the training set using one positive and one negative chain, instead of drawing and masking nodes at random from several chains.

We allow ourselves these approximations because our goal is not to compare accuracy values with the initial experiment. Instead, we seek to measure training times for two variants of implicit differentiation on a simple but representative toy example. One variant uses reverse-mode VJPs inside an iterative solve, with `Zygote.jl` (Innes, 2019) as the backend, while the other computes forward-mode sparse Jacobians for a direct solve, with `ForwardDiff.jl` (Revels et al., 2016) as the backend. The results are presented on Figure 4, where each dot corresponds to one training epoch. As we can see, the orange and blue curves follow nearly identical trajectories, with the exact same sequence of jumps and drops. This proves that for our setting, the direct method yields the same training results as the iterative one, which is not surprising when the linear system is non-degenerate. However, the orange curve is nearly two times faster in terms of time per epoch. Even though it starts with a slight delay (due to the initial sparsity detection), the direct

method based on sparse Jacobian matrices quickly catches up with and overtakes the iterative method based on VJPs.

Thus, sparse Jacobians can yield a sizeable training speedup. Here, part of it may be due to the number of matrix-vector product evaluations, and part of it to the increased efficiency of forward mode compared to reverse mode. The achievable speedup also depends on the parameters of the implicit linear solver, for instance the required precision or the maximum number of iterations.

In this specific instance, the sparsity pattern would have been straightforward to guess manually: it is a block version of the chain graph's adjacency matrix. However, realistic GNN architectures are much more convoluted: they may involve various feature channels, skip connections, graph rewiring, etc. With ASD, these complications are no longer a hurdle to the user, who can freely compute sparse Jacobians whenever implicit differentiation benefits from them.

## 6 Conclusion

ASD is an essential part of the scientific computing toolkit. While its core ideas have been known for decades, its adoption in high-level ML frameworks is still lagging. We presented a refreshed formulation of sparsity detection using operator overloading, and described an efficient software implementation which is already used at scale. Our hope is that such advances can spark renewed interest in sparse differentiation in the ML community and enable the practical use of Jacobian and Hessian matrices in domains where they were previously considered too expensive to compute.

Still, there are numerous research avenues to pursue. On the theoretical side, sparsity detection could be generalized to encompass various kinds of structures and symmetries, for instance block structure. This in turn could lead to efficient decomposition techniques for large-scale problems in a variety of domains. On the practical side, our operator overloading implementation only supports a limited amount of control flow when computing global sparsity patterns, for which some amount of program transformation is needed. The packages we developed were designed for CPUs, but deep learning applications will require GPU support and allocation-free routines, leveraging hardware-specific primitives. Finally, we plan to explore interoperability or adaptation for the Python language, which is the default choice in modern ML workflows.

**Acknowledgments**

We would like to thank Alexis Montoison for his valuable feedback and his work on `SparseMatrixColorings.jl`. We also thank Klaus-Robert Müller, Andreas Ziehe, Niklas Schmitz, Stefan Gugler and Marcel Langer for insightful discussions and feedback. Furthermore, we gratefully acknowledge funding from the German Federal Ministry of Education and Research under the grant BIFOLD25B.

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

## A    More applications

**Probabilistic modeling** makes frequent use of Hessian matrices. In frequentist statistics, the Fisher information matrix (Fisher, 1925) is defined as the expected Hessian of the negative log-likelihood: $\mathcal{I}(\boldsymbol{\theta}) = -\mathbb{E}_{\mathbf{x} \sim p(\cdot|\boldsymbol{\theta})}[\nabla^2_{\boldsymbol{\theta}} \log p(\mathbf{x}|\boldsymbol{\theta})]$. Its inverse gives an estimate of the variance for asymptotically Gaussian estimators. The Bayesian counterpart of this notion is Laplace approximation, whereby the posterior distribution of an estimator is approximated with a Gaussian. The precision of that Gaussian distribution is taken as the observed Fisher information. When the exact Hessian of the log-density is intractable to compute or invert, diagonal approximations are a common workaround. As we witness a shift from simple models to full-fledged probabilistic programs (van de Meent et al., 2021), AD becomes a key requirement to handle inscrutable log-density functions. Moreover, Markov-Chain Monte-Carlo is a family of techniques that allow sampling from high-dimensional, unnormalized densities (Robert and Casella, 2005). To better exploit the geometry of a distribution, Hamiltonian Monte-Carlo (Betancourt, 2018) incorporates derivatives in the simulation, and those derivatives can be computed with AD. Its Riemannian extension (Girolami and Calderhead, 2011) gives a central role to the Fisher information matrix when defining the metric tensor.

## B    ASD implementations in high-level programming languages

Among scientific computing languages, ASD has percolated most prominently into the MATLAB ecosystem, but the closed-source nature of the language hinders adoption in ML communities. As for Python and R, their ASD libraries are currently less developed than their Julia counterparts.

**MATLAB.** The `ADMIT` package (Coleman and Verma, 2000) relies on an external (C or MATLAB) AD tool to compute sparse Jacobians and Hessians, augmenting it with coloring and compression. However, the chosen AD tool must also be able to detect Jacobian and Hessian sparsity patterns, as is the case for `ADMAT` (Coleman and Verma, 1998) and `ADOL-C` (Griewank et al., 1996). `ADiMAT` (Willkomm et al., 2014) has similar sparse functionality but requires user input for the sparsity pattern. The `MAD` (Forth, 2006) library offers two options for sparse Jacobians and Hessians: either coloring and compression, or direct use of sparse derivative storage inside elementary operations. `MSAD` (Kharche and Forth, 2006) enhances `MAD` by replacing operator overloading with source transformation. Finally, `ADiGator` (Weinstein and Rao, 2017) moves as much complexity as possible to compile time, in order to lessen the runtime impact of ASD.

**Python.** The main AD frameworks in Python are `TensorFlow` (Abadi et al., 2015), `PyTorch` (Paszke et al., 2019) and `JAX` (Bradbury et al., 2018). We are not aware of any ASD libraries for `Tensorflow` or `PyTorch`. In `JAX`, we mostly found `sparsejac` (Schubert, 2024), which is limited to Jacobians and does not support sparsity detection. A blog post by Simpson (2024) describes potential avenues for sparsity detection in `JAX`, but they are not fully implemented. The `Graphax` package (Lohoff and Neftci, 2024) implements the *cross-country elimination* technique described in subsection 2.4. Since it can exploit sparsity in Jacobian computations, it could potentially be used as an alternative to ASD when vectorized code is involved. Finally, the library `auto_diff` (Nobel, 2020) computes sparse Jacobians of plain `NumPy` code.

**R.** Our search for ASD in R turned up two packages: `sparseHessianFD` Braun (2017), which asks the user to provide a gradient, and `TMB` (Kristensen et al., 2016), which is focused on one statistical application (Laplace approximation) and requires some of the code to be written in C++.

**Julia.** The state of the art for ASD used to involve a combination of `Symbolics.jl` (Gowda et al., 2019; 2022) with `SparseDiffTools.jl` (JuliaDiff contributors, 2024). It was limited to a couple of AD backends, not optimized to support Hessian matrices, and (as our experiments show) rather slow for sparsity detection. As a result, the ecosystem is currently switching to a new, more modular pipeline based on `DifferentiationInterface.jl` (Dalle and Hill, 2025b) and `SparseMatrixColorings.jl` (Dalle and Montoison, 2025).

**Remark 3.** *During preparation of this paper, we became aware of ongoing work around* `Spadina` *(Moses, 2023), which relies on dead code removal during compilation to compute only nonzero matrix entries. It is planned to interface with* `Enzyme` *(Moses and Churavy, 2020; Moses et al., 2021) and* `JAX`.

## C Coloring basics

Here we give a brief primer on coloring of sparse matrices for ASD, as implemented in `SparseMatrixColorings.jl`. The goal is to make this paper self-contained, but we refer the reader to Montoison et al. (2025) for more details. In our experiments, we only use greedy column coloring with direct decompression (see below).

### C.1 Compression and decompression

As explained in the review by Gebremedhin et al. (2005), coloring is a necessary step for matrix *compression*. Consider a matrix $\boldsymbol{A} \in \mathbb{R}^{m \times n}$ and a mapping from columns to colors $\phi : \{1, \ldots, n\} \to \{1, \ldots, c_n\}$, where $c_n$ is the largest color used. We construct the compressed matrix $\boldsymbol{B} \in \mathbb{R}^{m \times c_n}$ such that each column of $\boldsymbol{B}$ corresponds to a color $k$. The column $\boldsymbol{B}_{:,k}$ is obtained by summing the columns of $\boldsymbol{A}$ which share that same color:

$$\boldsymbol{B}_{:,k} = \sum_{\substack{1 \le j \le n \\ \phi(j)=k}} \boldsymbol{A}_{:,j} = \boldsymbol{A}\boldsymbol{v}^{(k)} \qquad \text{with} \qquad \boldsymbol{v}^{(k)} = \sum_{\substack{1 \le j \le n \\ \phi(j)=k}} \boldsymbol{e}^{(j)}.$$

If the coloring $\phi$ induces an injective mapping $\boldsymbol{A} \mapsto \boldsymbol{B}$, then *decompression* is possible: $\boldsymbol{B}$ contains all the necessary information to recover $\boldsymbol{A}$ exactly. Thus, it is possible to evaluate $\boldsymbol{A}$ using only $c_n < n$ matrix-vector products: one for each multi-basis vector $\boldsymbol{v}^{(k)}$.

In the general case, decompression from $\boldsymbol{B}$ to $\boldsymbol{A}$ might require solving a linear system. To avoid paying this cost, it common to consider a stronger condition called *direct* decompression: every coefficient $\boldsymbol{A}_{i,j}$ should be directly readable from some coefficient $\boldsymbol{B}_{l,k}$. The question then becomes: what constraints should we impose on the coloring $\phi$ to ensure that direct decompression is possible?

### C.2 Coloring asymmetric matrices

The following explanation is summarized from Gebremedhin et al. (2005, Section 3). In the asymmetric case, direct decompression is possible if $\phi$ induces a *structurally orthogonal* partition of the columns of $\boldsymbol{A}$: for every non-zero $\boldsymbol{A}_{i,j}$, column $j$ should be the only one with a non-zero in row $i$ among its color group. With such a partition, when we sum the columns inside the group $\phi(j)$, no other entry is added to $\boldsymbol{A}_{i,j}$, which implies that $\boldsymbol{B}_{i,\phi(j)} = \boldsymbol{A}_{i,j}$.

Let us represent the matrix $\boldsymbol{A}$ with a *bipartite graph* $\mathcal{G}_b = (\mathcal{I} \cup \mathcal{J}, \mathcal{E}_b)$. Its vertices include the set of rows $\mathcal{I} = \{1, \ldots m\}$ and the set of columns $\mathcal{J} = \{1, \ldots, n\}$, while its edges link rows and columns which intersect at a non-zero element: $\mathcal{E}_b = \{(i,j) \in \mathcal{I} \times \mathcal{J} : \boldsymbol{A}_{i,j} \ne 0\}$. The structural orthogonality condition is satisfied whenever $\phi$ is a *partial distance-2 coloring* of the column vertices $\mathcal{J}$ in $\mathcal{G}_b$. In other words, $\phi$ should assign a color to each column such that if $j_1$ and $j_2$ are distance-2 neighbors in the bipartite graph (there exists $i$ such that $(i,j_1) \in \mathcal{E}_b$ and $(i,j_2) \in \mathcal{E}_b$), then they have different colors $\phi(j_1) \ne \phi(j_2)$.

To achieve this result, a simple greedy algorithm suffices: iterate over columns in their native order, and give each one the smallest color that is still not used among its distance-2 neighbors. We use this very algorithm in our Jacobian computation experiments.

### C.3 Coloring symmetric matrices

The following explanation is summarized from Gebremedhin et al. (2005, Section 4). In the symmetric case, direct decompression is possible if $\phi$ induces a *symmetrically orthogonal* partition of the columns of $\boldsymbol{A}$ : for every non-zero $\boldsymbol{A}_{i,j}$, either column $j$ should be the only one with a non-zero in row $i$ among its color group, or column $i$ should be the only one with a non-zero in row $j$ among its color group. This is a more relaxed notion of partition, which makes sense because we have more opportunities for recovery: the coefficient $\boldsymbol{A}_{i,j} = \boldsymbol{A}_{i,j}$ can be read either from $\boldsymbol{B}_{i,\phi(j)}$ or from $\boldsymbol{B}_{j,\phi(i)}$.

Let us represent the matrix $\boldsymbol{A}$ with an *adjacency graph* $\mathcal{G}_a = (\mathcal{J}, \mathcal{E}_a)$, whose vertices are the set of columns (or rows) and whose edges are defined by $\mathcal{E}_a = \{(i,j) \in \mathcal{J}^2 : \boldsymbol{A}_{i,j} \ne 0\}$. The symmetric orthogonality

condition is satisfied whenever $\phi$ is a *star coloring* of $\mathcal{G}_a$. A star coloring is a standard (distance-1) coloring with the additional constraint that every path on four vertices uses at least three colors. In a star coloring, every subgraph induced by a pair of colors is a collection of stars.

Again, a simple greedy algorithm suffices: iterate over columns in their native order, and give each one the smallest color that is still not used among its neighbors, while avoiding the creation of four-vertex three-colored paths. As observed by Gebremedhin et al. (2007), representing the graph as a set of two-colored stars provides an efficient implementation of this idea. We use an optimized version of their algorithm in our Hessian computation experiments.

### C.4   Other details

Both asymmetric and symmetric decompression routines are optimized for Julia's default sparse matrix format, Compressed Sparse Column (CSC).

## D   Hessian tracing

Here we extend the reasoning of section 3 to second-order cross-derivatives.

### D.1   Principle

Let $f\colon \mathbb{R}^n \to \mathbb{R}$ be a vector-to-scalar function, and $\boldsymbol{x} \in \mathbb{R}^n$. This time, we want the sparsity pattern of the Hessian matrix $\nabla^2 f(\boldsymbol{x}) \in \mathbb{R}^{n\times n}$. Extending what we did for gradients, we define a second-order tracer type which contains a primal value $y(\boldsymbol{x}) \in \mathbb{R}$, its gradient sparsity pattern $\mathbf{1}[\nabla y(\boldsymbol{x})] \in \{0,1\}^n$ and its Hessian sparsity pattern $\mathbf{1}[\nabla^2 y(\boldsymbol{x})] \in \{0,1\}^{n\times n}$. Again, the Hessian in question is taken with respect to the input $\boldsymbol{x}$, and we will replace every intermediate scalar quantity in $F$ with a tracer. We start by turning every $x_j$ into $(x_j, \boldsymbol{e}_j, \boldsymbol{0})$ (where the third term is the initial empty Hessian sparsity pattern), and at the end we recover $(F(\boldsymbol{x}), \mathbf{1}[\nabla F(\boldsymbol{x})], \mathbf{1}[\nabla^2 F(\boldsymbol{x})])$. This time, we need to describe how Hessian sparsity patterns propagate.

**Remark 4.** *Our second-order tracer is related to hyperdual numbers, which can be found in second-order forward-mode AD (Fike and Alonso, 2012). More precisely, it describes the sparsity pattern of a batched hyperdual number, which could be used to implement second-order batched forward-mode AD. One can also see it as the binary version of the sparse triplet in Dixon et al. (1990).*

### D.2   Second-order propagation rules

We reuse the same framework as for Jacobian tracing in subsection 3.2, but this time we go one step further. Differentiating Equation 1 once more with respect to $x_i$ gives us:

$$\partial_{ij}^2 \gamma = \partial_i \partial_j \gamma = d_{x_i}\left[\partial_1 \varphi \cdot \partial_j \alpha\right] + d_{x_i}\left[\partial_2 \varphi \cdot \partial_j \beta\right]$$

Now we apply the product rule:

$$\partial_{ij}^2 \gamma = \left[d_{x_i}\partial_1 \varphi \cdot \partial_j \alpha + \partial_1 \varphi \cdot \partial_i \partial_j \alpha\right] + \left[d_{x_i}\partial_2 \varphi \cdot \partial_j \beta + \partial_2 \varphi \cdot \partial_i \partial_j \beta\right]$$

We recognize second-order derivatives in the second term of each bracket:

$$\partial_{ij}^2 \gamma = \left[d_{x_i}\left(\partial_1 \varphi\right) \cdot \partial_j \alpha + \partial_1 \varphi \cdot \partial_{ij}^2 \alpha\right] + \left[d_{x_i}\left(\partial_2 \varphi\right) \cdot \partial_j \beta + \partial_2 \varphi \cdot \partial_{ij}^2 \beta\right] \tag{3}$$

For the first term of each bracket, we can once again apply Equation 1 but with the differentiated operators $\partial_1 \varphi$ and $\partial_2 \varphi$ instead of $\varphi$, and with the total derivative $d_{x_i}$ instead of $d_{x_j}$:

$$d_{x_i}\left(\partial_1 \varphi\right) = \partial_1\left(\partial_1 \varphi\right) \cdot \partial_i \alpha + \partial_2\left(\partial_1 \varphi\right) \cdot \partial_i \beta$$
$$= \partial_1^2 \varphi \cdot \partial_i \alpha + \partial_{12}^2 \varphi \cdot \partial_i \beta$$
$$d_{x_i}\left(\partial_2 \varphi\right) = \partial_1\left(\partial_2 \varphi\right) \cdot \partial_i \alpha + \partial_2\left(\partial_2 \varphi\right) \cdot \partial_i \beta$$
$$= \partial_{12}^2 \varphi \cdot \partial_i \alpha + \partial_2^2 \varphi \cdot \partial_i \beta$$

Plugging these into Equation 3, we get:

$$\partial_{ij}^2\gamma = \quad \left[(\partial_1^2\varphi \cdot \partial_i\alpha + \partial_{12}^2\varphi \cdot \partial_i\beta) \cdot \partial_j\alpha + \partial_1\varphi \cdot \partial_{ij}^2\alpha\right]$$
$$+ \left[(\partial_{12}^2\varphi \cdot \partial_i\alpha + \partial_2^2\varphi \cdot \partial_i\beta) \cdot \partial_j\beta + \partial_2\varphi \cdot \partial_{ij}^2\beta\right]$$

And sorting by the operator derivatives involved, we conclude:

$$\partial_{ij}^2\gamma = \quad \partial_1\varphi \cdot \partial_{ij}^2\alpha + \partial_2\varphi \cdot \partial_{ij}^2\beta \qquad \text{(first derivatives)}$$
$$+ \partial_1^2\varphi \cdot \partial_i\alpha \cdot \partial_j\alpha + \partial_2^2\varphi \cdot \partial_i\beta \cdot \partial_j\beta \qquad \text{(second derivatives)}$$
$$+ \partial_{12}^2\varphi \cdot \partial_i\alpha \cdot \partial_j\beta + \partial_{12}^2\varphi \cdot \partial_i\beta \cdot \partial_j\alpha \qquad \text{(cross derivatives)}$$

Bringing the indices together with vector and matrix notation shows us:

$$\nabla^2\gamma = \quad \partial_1\varphi \cdot \nabla^2\alpha + \partial_2\varphi \cdot \nabla^2\beta$$
$$+ \partial_1^2\varphi \cdot (\nabla\alpha)(\nabla\alpha)^\top + \partial_2^2\varphi \cdot (\nabla\beta)(\nabla\beta)^\top$$
$$+ \partial_{12}^2\varphi \cdot (\nabla\alpha)(\nabla\beta)^\top + \partial_{12}^2\varphi \cdot (\nabla\beta)(\nabla\alpha)^\top$$

And once again, the sparsity pattern emerges, using $\vee$ for elementwise OR between two matrices. Like before, we use $\leq$ to emphasize the fact that the resulting sparsity patterns are overestimates, which may be too conservative.

$$\mathbf{1}[\nabla^2\gamma] \leq \left|\begin{array}{lll} & \mathbf{1}[\partial_1\varphi] \cdot \mathbf{1}[\nabla^2\alpha] & \vee \quad \mathbf{1}[\partial_2\varphi] \cdot \mathbf{1}[\nabla^2\beta] \\ \vee & \mathbf{1}[\partial_1^2\varphi] \cdot \mathbf{1}[(\nabla\alpha)(\nabla\alpha)^\top] & \vee \quad \mathbf{1}[\partial_2^2\varphi] \cdot \mathbf{1}[(\nabla\beta)(\nabla\beta)^\top] \\ \vee & \mathbf{1}[\partial_{12}^2\varphi] \cdot \mathbf{1}[(\nabla\alpha)(\nabla\beta)^\top] & \vee \quad \mathbf{1}[\partial_{12}^2\varphi] \cdot \mathbf{1}[(\nabla\beta)(\nabla\alpha)^\top] \end{array}\right.$$

Let us generalize $\vee$ to also represent the outer product OR between two vectors, so that $\mathbf{1}[\boldsymbol{ab}^\top] = \mathbf{1}[\boldsymbol{a}] \vee \mathbf{1}[\boldsymbol{b}]^\top$. This gives our final expression:

$$\mathbf{1}[\nabla^2\gamma] \leq \left|\begin{array}{lll} & \mathbf{1}[\partial_1\varphi] \cdot \mathbf{1}[\nabla^2\alpha] & \vee \quad \mathbf{1}[\partial_2\varphi] \cdot \mathbf{1}[\nabla^2\beta] \\ \vee & \mathbf{1}[\partial_1^2\varphi] \cdot (\mathbf{1}[\nabla\alpha] \vee \mathbf{1}[\nabla\alpha]^\top) & \vee \quad \mathbf{1}[\partial_2^2\varphi] \cdot (\mathbf{1}[\nabla\beta] \vee \mathbf{1}[\nabla\beta]^\top) \\ \vee & \mathbf{1}[\partial_{12}^2\varphi] \cdot (\mathbf{1}[\nabla\alpha] \vee \mathbf{1}[\nabla\beta]^\top) & \vee \quad \mathbf{1}[\partial_{12}^2\varphi] \cdot (\mathbf{1}[\nabla\beta] \vee \mathbf{1}[\nabla\alpha]^\top) \end{array}\right. \tag{4}$$

As we can see, the propagation of Hessian sparsity patterns through the operator $\varphi$ only depends on five values:

$$\mathbf{1}[\partial_1\varphi], \quad \mathbf{1}[\partial_2\varphi], \quad \mathbf{1}[\partial_1^2\varphi], \quad \mathbf{1}[\partial_2^2\varphi] \quad \text{and} \quad \mathbf{1}[\partial_{12}^2\varphi]$$

These binary values tell us whether the operator $\varphi$ locally depends on each of its arguments at the first and second order.

Thanks to operator overloading at the scalar level, if the control flow reaches a dead end (a value which is not reused for the function output), the corresponding dependencies will not appear in the computed sparsity pattern. This contrasts with the method of (Walther, 2008), where all intermediate values contribute to the final result, leading to potential overestimation of the Hessian sparsity pattern.

### D.3 Second-order operator classification

To implement Equation 4, we need to refine the classification from Table 1 by considering second derivatives. Once again, the distinction between local and global sparsity plays a key role. Some examples are given in Table 3.

## E Tensor-level overload example

Using the standard matrix multiplication algorithm, propagating tracers through the matrix multiplication $\boldsymbol{C} = \boldsymbol{AB}$ with $\boldsymbol{A} \in \mathbb{R}^{n \times p}$ and $\boldsymbol{B} \in \mathbb{R}^{p \times m}$ requires $n \cdot m \cdot p$ multiplications and $n \cdot m \cdot (p-1)$ additions, since for all $n \cdot m$ entries $C_{i,j}$,

$$C_{i,j} = \sum_{k=1}^p A_{i,k}B_{k,j}. \tag{5}$$

| Operator $\varphi(\alpha, \beta)$ | Local | | | Global | | |
|---|---|---|---|---|---|---|
| | $\mathbf{1}[\partial_1^2\varphi]$ | $\mathbf{1}[\partial_2^2\varphi]$ | $\mathbf{1}[\partial_{12}^2\varphi]$ | $\mathbf{1}[\partial_1^2\varphi]$ | $\mathbf{1}[\partial_2^2\varphi]$ | $\mathbf{1}[\partial_{12}^2\varphi]$ |
| exp, log | 1 | – | – | 1 | – | – |
| +, –, max, min | 0 | 0 | 0 | 0 | 0 | 0 |
| $\star$ | 0 | 0 | 1 a.e. | 0 | 0 | 1 |
| / | 0 | 1 a.e. | 1 a.e. | 0 | 1 | 1 |

Table 3: Second-order classification of operators
Unary operators have no second argument. "a.e." means "almost everywhere" for the Lebesgue measure

Applying the first-order propagation rule from Equation 2 to the scalar multiplication operator

$$\gamma(\boldsymbol{x}) = \varphi(x_1, x_2) = x_1 x_2 \,,$$

we obtain the global propagation rule

$$\mathbf{1}[\nabla\gamma] = \mathbf{1}[\partial_1\varphi] \cdot \mathbf{1}[\nabla x_1] \vee \mathbf{1}[\partial_2\varphi] \cdot \mathbf{1}[\nabla x_2] = \mathbf{1}[\nabla x_1] \vee \mathbf{1}[\nabla x_2] \,.$$

An identical propagation rule can also be derived for addition. Inserting into Equation 5, we obtain

$$\mathbf{1}[\nabla C_{i,j}] = \bigvee_{k=1}^{p} \mathbf{1}[\nabla A_{i,k}] \vee \mathbf{1}[\nabla B_{k,j}] \,,$$

which, if naively implemented, requires a total of $n \cdot m \cdot (2p - 1)$ elementwise OR operations to propagate tracers through the entire matrix multiplication. Rewriting this as the equivalent

$$\mathbf{1}[\nabla C_{i,j}] = \underbrace{\bigvee_{k=1}^{p} \mathbf{1}[\nabla A_{i,k}]}_{=:\mathbf{1}[\nabla \bar{A}_i]} \vee \underbrace{\bigvee_{k=1}^{p} \mathbf{1}[\nabla B_{k,j}]}_{=:\mathbf{1}[\nabla \bar{B}_j]}$$

reveals that we can instead first compute intermediate quantities $\mathbf{1}[\nabla \bar{A}_i]$ and $\mathbf{1}[\nabla \bar{B}_j]$ by taking $n \cdot (p - 1)$ elementwise OR operations across rows of $\boldsymbol{A}$ and $m \cdot (p-1)$ operations across columns of $\boldsymbol{B}$ respectively. The total amount of operations is therefore reduced to $(n + m)(p - 1) + n \cdot m$, leading to a significant increase in performance.

## F  Code demonstration

We now showcase the API of the ASD pipeline used in our experiments section 5.

### F.1  Sparsity detection

To demonstrate the generality of SCT's tracer-based approach to sparsity detection, we compute the global Jacobian sparsity pattern of a convolutional layer provided by the deep learning framework Flux.jl (Innes et al., 2018; Innes, 2018).

```julia
using SparseConnectivityTracer, Flux   # import required packages

x = randn(Float32, 10, 10, 3, 1)       # create input tensor
layer = Conv((5, 5), 3 => 1)           # create convolutional layer

detector = TracerSparsityDetector()    # specify global sparsity pattern
jacobian_sparsity(layer, x, detector)  # compute pattern
```

Listing 1: Detecting the Jacobian sparsity pattern of a convolutional layer using
SparseConnectivityTracer.jl

The full code is shown in Listing 1. We import the two required packages and sample a random input tensor $x$ in the size of a $10 \times 10$ image with 3 color channels and a batch size of 1. We then create a convolutional layer `Conv` with a kernel of size $5 \times 5$, mapping the 3 input channels to a single output channel. To compute global sparsity patterns, `TracerSparsityDetector` is chosen[10]. The Jacobian sparsity pattern is then computed by calling the `jacobian_sparsity` function. Note that SCT doesn't implement custom overloads for convolutional layers. Instead, `Flux.jl`'s generic implementation of a convolution falls back to elementary operators like addition and multiplication, which are overloaded on SCT's tracer types. This is enabled by Julia's *multiple dispatch* paradigm and doesn't require writing any additional code.

The resulting sparsity pattern is shown in Figure 5a, with colors resulting from subsequent greedy column coloring. The banded structure of the matrix results from the size of the convolutional kernel as well as the number of input channels. Figure 5c shows the sparsity pattern that results from increasing the batch size from 1 to 2. Since the convolutions of the two inputs in the batch are parallel and separable computations, the resulting sparsity pattern is a block diagonal matrix. Since this is the case for all separable parallel computations, sparsity detection can be used as a tool to debug the automatic parallelization of computer programs. Figure 5d additionally increases the number of output channels from 1 to 2. Note that while the size of the Jacobian sparsity pattern increases across all three figures, the number of of colors stays constant.

## F.2 Computing Jacobians with AD and ASD

Listings 2 and 3 both show the computation of the Jacobian of a convolutional layer from `Flux.jl`, the former using AD, the latter using ASD. Since ASD is fully automatic, the only difference in code between the two computations lies in the package imports and the specification of the backend used to call to the `jacobian` function. Possible choices include `ForwardDiff.jl` (Revels et al., 2016), `ReverseDiff.jl` (Revels, 2016), `Zygote.jl` (Innes, 2019; Innes et al., 2019) and `Enzyme.jl` (Moses and Churavy, 2020; Moses et al., 2021).

```
# Import required packages
using DifferentiationInterface  # common interface to AD backends
using ForwardDiff               # forward-mode AD backend
using Flux                      # deep learning framework

# Create input tensor and convolutional layer
x = randn(Float32, 10, 10, 3, 1)
layer = Conv((5, 5), 3 => 1)

# Specify AD backend
ad_backend = AutoForwardDiff()

# Compute Jacobian
jacobian(layer, ad_backend, x)
```

Listing 2: AD computation of the Jacobian of a convolutional layer using `DifferentiationInterface.jl`

```
# Import required packages
using SparseConnectivityTracer  # sparsity detection
using SparseMatrixColorings     # sparsity pattern coloring
using DifferentiationInterface  # common interface to AD backends
using ForwardDiff               # forward-mode AD backend
using Flux                      # deep learning framework

# Create input tensor and convolutional layer
x = randn(Float32, 10, 10, 3, 1)
layer = Conv((5, 5), 3 => 1)

# Specify ASD backend
asd_backend = AutoSparse(
    AutoForwardDiff();
    sparsity_detector=TracerSparsityDetector(), # from SparseConnectivityTracer
    coloring_algorithm=GreedyColoringAlgorithm(), # from SparseMatrixColorings
```

---

[10]For local sparsity detection, `TracerLocalSparsityDetector` is used instead of `TracerSparsityDetector`.

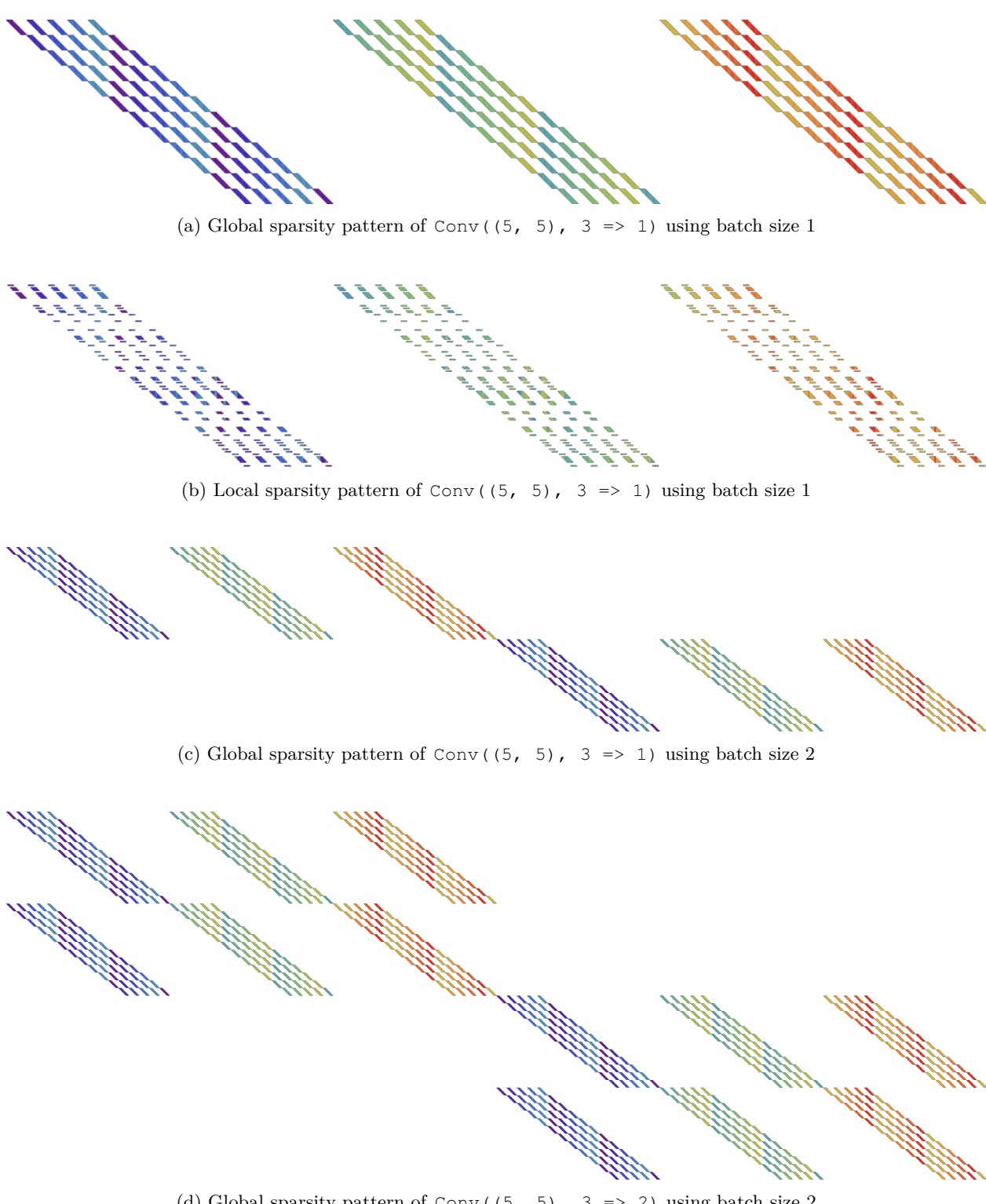

(a) Global sparsity pattern of `Conv((5, 5), 3 => 1)` using batch size 1

(b) Local sparsity pattern of `Conv((5, 5), 3 => 1)` using batch size 1

(c) Global sparsity pattern of `Conv((5, 5), 3 => 1)` using batch size 2

(d) Global sparsity pattern of `Conv((5, 5), 3 => 2)` using batch size 2

Figure 5: Jacobian sparsity patterns of small convolutional layers from `Flux.jl` applied to a $10 \times 10$ image with three color channels. Squares correspond to non-zero entries in the Jacobian, with colors resulting from greedy column coloring.

```
)

# Compute Jacobian
jacobian(layer, asd_backend, x)
```

Listing 3: ASD computation of the Jacobian of a convolutional layer using
`DifferentiationInterface.jl`, `SparseConnectivityTracer.jl` and
`SparseMatrixColorings.jl`

## G   More Jacobian experiments: Brusselator

### G.1   Sparsity detection

In Figure 6, we display the colored sparsity patterns of the Brusselator PDE (see section 5) for various discretization levels. Writing explicit formulas for this kind of intricate structure is fairly tiresome and subject to human error, which justifies automated sparsity detection approaches.

### G.2   Jacobian computation and linear solves

In Table 4, we present two sets of results:

1. A comparison between AD and ASD for Jacobian computation, already discussed in section 5.

2. A comparison between various methods for solving a Jacobian linear system, which we now focus on. Solving linear systems involving Jacobian matrices is typically useful as part of a Newton root-finding step. This can happen within a larger iteration, like a backward Euler method for differential equations.

We distinguish between three different techniques for the Newton step. First, one needs to choose between direct solvers (which factorize the matrix) and iterative solvers. Second, if one picks an iterative solver, there is still a choice between precomputing the matrix or using the lazy JVP operator. Our specific implementation uses Julia's standard library `SparseArrays.jl` for the direct solve, which in turn relies on `SuiteSparse`. Meanwhile, the iterative solver is the default recommendation `BiCGStab(l)` from `IterativeSolvers.jl` (Chen et al., 2013). In the rightmost part of Table 4, we observe that the lazy version wins for small instances, but then the direct solver is actually faster for large enough matrices.

| Problem | Jacobian computation[1] | | | | | Newton step[1] | | |
|---|---|---|---|---|---|---|---|---|
| N | AD (prepared) | ASD (prepared) | [2] | ASD (unprepared) | [2] | JVP (iterative) | Jacobian (iterative) | Jacobian (direct) |
| 6 | $1.64 \cdot 10^{-5}$ | $\mathbf{1.97 \cdot 10^{-6}}$ | $\mathbf{(8.3)}$ | $3.36 \cdot 10^{-5}$ | $(0.5)$ | $\mathbf{2.07 \cdot 10^{-5}}$ | $2.19 \cdot 10^{-5}$ | $4.52 \cdot 10^{-5}$ |
| 12 | $2.44 \cdot 10^{-4}$ | $\mathbf{8.67 \cdot 10^{-6}}$ | $\mathbf{(28.1)}$ | $1.70 \cdot 10^{-4}$ | $(1.4)$ | $\mathbf{1.34 \cdot 10^{-4}}$ | $1.61 \cdot 10^{-4}$ | $2.42 \cdot 10^{-4}$ |
| 24 | $4.02 \cdot 10^{-3}$ | $\mathbf{3.43 \cdot 10^{-5}}$ | $\mathbf{(117.2)}$ | $1.31 \cdot 10^{-3}$ | $(3.1)$ | $\mathbf{1.04 \cdot 10^{-3}}$ | $1.34 \cdot 10^{-3}$ | $1.24 \cdot 10^{-3}$ |
| 48 | $7.60 \cdot 10^{-2}$ | $\mathbf{1.68 \cdot 10^{-4}}$ | $\mathbf{(451.4)}$ | $1.70 \cdot 10^{-2}$ | $(4.5)$ | $\mathbf{8.34 \cdot 10^{-3}}$ | $1.17 \cdot 10^{-2}$ | $8.98 \cdot 10^{-3}$ |
| 96 | $1.35 \cdot 10^{0}$ | $\mathbf{6.68 \cdot 10^{-4}}$ | $\mathbf{(2017.2)}$ | $2.16 \cdot 10^{-1}$ | $(6.2)$ | $7.56 \cdot 10^{-2}$ | $1.11 \cdot 10^{-1}$ | $\mathbf{4.07 \cdot 10^{-2}}$ |
| 192 | $2.25 \cdot 10^{1}$ | $\mathbf{3.09 \cdot 10^{-3}}$ | $\mathbf{(7293.5)}$ | $4.62 \cdot 10^{0}$ | $(4.9)$ | $1.05 \cdot 10^{0}$ | $1.07 \cdot 10^{0}$ | $\mathbf{2.30 \cdot 10^{-1}}$ |

[1] Wall time in seconds.
[2] In parentheses: Wall time ratio compared to prepared AD (higher is better).

Table 4: Performance comparison of Jacobian computation & linear solves on the Brusselator PDE.

While this behavior might seem surprising, it is easily explained by the principles underlying ASD and iterative solvers. Assuming that the function at hand is very expensive to compute, pure linear-algebraic manipulations will be much cheaper in comparison, even sophisticated ones like `SuiteSparse`'s linear solve.

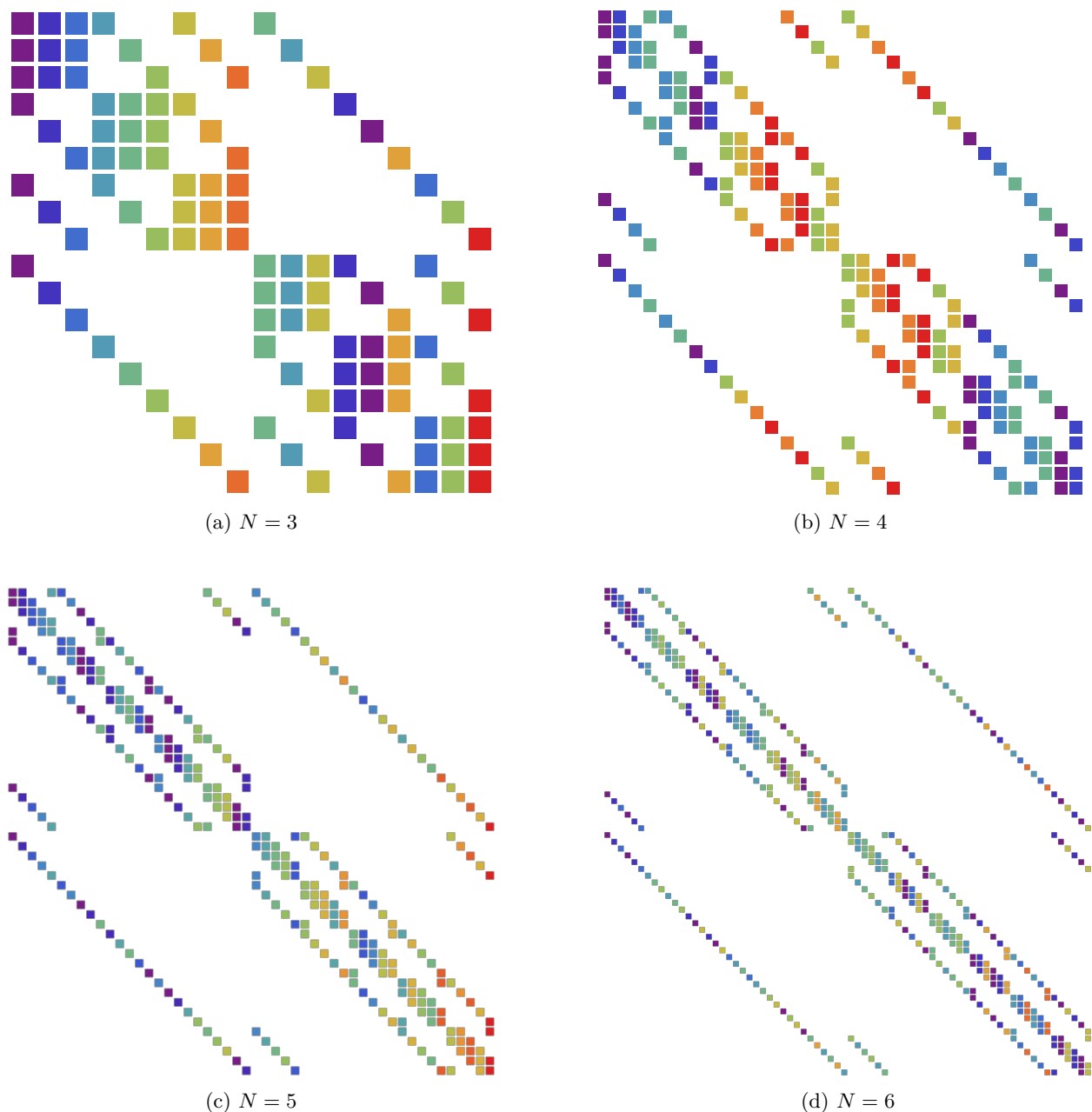

(a) $N = 3$

(b) $N = 4$

(c) $N = 5$

(d) $N = 6$

Figure 6: Jacobian sparsity patterns of the discretized Brusselator PDE of size $N \times N \times 2$. Squares correspond to non-zero entries in the Jacobian, with colors resulting from greedy column coloring.

Thus, it all comes down to how many JVPs need to be performed. Iterative solvers require one JVP per iteration, so the total complexity is dictated by the precision requirement and the conditioning number of the Jacobian. Meanwhile, materializing a Jacobian with ASD requires one JVP per distinct color, so the total complexity is dictated by the sparsity pattern only. Thus, in cases where high precision is required, the matrix is ill-conditioned and the coloring number is low, ASD with a direct solver can prevail over its lazy iterative counterpart.

## H Hessian experiments: ACOPF

### H.1 Sparsity detection

At the core of power systems planning and power markets lies a nonlinear constrained optimization problem called the *alternating current optimal power flow* (ACOPF) problem. As of today, the full ACOPF remains unsolved. While approximate solution techniques exist, they result in unnecessary emissions and spending, with potential annual savings from an optimal solution estimated in the tens of billions of dollars (Cain et al., 2012). The problem has also attracted the interest of the ML community with recent developments in neural ACOPF solvers (Piloto et al., 2024). To validate algorithms for the OPF problem, a suite of benchmarks called *Power Grid Lib* (PGLib) has been developed by Babaeinejadsarookolaee et al. (2021), which can be run via the `rosetta-opf` (Coffrin and Dowson, 2022) implementation of the ACOPF.

We benchmark the sparsity detection of `Symbolics.jl` and SCT on the Hessian of the Lagrangian of several PGLib optimization problems. The results are summarized in Table 5. SCT outperforms `Symbolics.jl` on every problem, regardless of size and sparsity.

### H.2 Hessian computation

We now benchmark the computation of dense and sparse Hessians for the Lagrangian of several PGLib optimization problems. Our methodology with respect to *prepared* and *unprepared* computations mirrors that used in subsubsection 5.1.2. In both AD and ASD benchmarks, `ForwardDiff.jl` is used over the reverse-mode backend `ReverseDiff.jl` to evaluate HVPs.

The results are summarized in Table 6. For large problems, prepared ASD provides an increase in performance of three orders of magnitude over AD. Even one-off unprepared ASD provides performance benefits over AD in all PGLib cases. Once again, performance gains of one-off ASD on small problems are largely due to the performance of SCT: timings of the *3_lmbd* problem in Table 6 reveal that just the sparsity detection of `Symbolics.jl` alone was previously less performant than the full computation of the Hessian with AD.

| Problem | | Sparsity | | Sparsity detection[1] | | |
|---|---|---|---|---|---|---|
| Name | Inputs | Zeros | Colors[2] | Symbolics | SCT[3] | |
| *3_lmbd* | 24 | 91.15% | 6 | $1.29 \cdot 10^{-3}$ | $\mathbf{5.59 \cdot 10^{-5}}$ | **(23.1)** |
| *5_pjm* | 44 | 94.99% | 8 | $2.49 \cdot 10^{-3}$ | $\mathbf{1.19 \cdot 10^{-4}}$ | **(20.9)** |
| *14_ieee* | 118 | 97.84% | 10 | $9.02 \cdot 10^{-3}$ | $\mathbf{5.19 \cdot 10^{-4}}$ | **(17.4)** |
| *24_ieee_rts* | 266 | 99.22% | 12 | $1.92 \cdot 10^{-2}$ | $\mathbf{1.50 \cdot 10^{-3}}$ | **(12.8)** |
| *30_as* | 236 | 98.89% | 12 | $2.04 \cdot 10^{-2}$ | $\mathbf{1.60 \cdot 10^{-3}}$ | **(12.7)** |
| *30_ieee* | 236 | 98.89% | 12 | $2.03 \cdot 10^{-2}$ | $\mathbf{1.60 \cdot 10^{-3}}$ | **(12.7)** |
| *39_epri* | 282 | 99.10% | 10 | $2.45 \cdot 10^{-2}$ | $\mathbf{2.10 \cdot 10^{-3}}$ | **(11.7)** |
| *57_ieee* | 448 | 99.41% | 14 | $4.68 \cdot 10^{-2}$ | $\mathbf{4.91 \cdot 10^{-3}}$ | **(9.5)** |
| *60_c* | 518 | 99.56% | 12 | $5.15 \cdot 10^{-2}$ | $\mathbf{5.76 \cdot 10^{-3}}$ | **(8.9)** |
| *73_ieee_rts* | 824 | 99.74% | 12 | $9.84 \cdot 10^{-2}$ | $\mathbf{1.09 \cdot 10^{-2}}$ | **(9.0)** |
| *89_pegase* | 1042 | 99.74% | 26 | $1.80 \cdot 10^{-1}$ | $\mathbf{2.20 \cdot 10^{-2}}$ | **(8.2)** |
| *118_ieee* | 1088 | 99.77% | 12 | $1.57 \cdot 10^{-1}$ | $\mathbf{2.11 \cdot 10^{-2}}$ | **(7.5)** |
| *162_ieee_dtc* | 1484 | 99.82% | 16 | $2.99 \cdot 10^{-1}$ | $\mathbf{3.33 \cdot 10^{-2}}$ | **(9.0)** |
| *179_goc* | 1468 | 99.83% | 14 | $2.59 \cdot 10^{-1}$ | $\mathbf{3.09 \cdot 10^{-2}}$ | **(8.4)** |
| *197_snem* | 1608 | 99.85% | 14 | $3.02 \cdot 10^{-1}$ | $\mathbf{3.57 \cdot 10^{-2}}$ | **(8.5)** |
| *200_activ* | 1456 | 99.82% | 12 | $2.59 \cdot 10^{-1}$ | $\mathbf{2.92 \cdot 10^{-2}}$ | **(8.9)** |
| *240_pserc* | 2558 | 99.91% | 16 | $6.72 \cdot 10^{-1}$ | $\mathbf{7.32 \cdot 10^{-2}}$ | **(9.2)** |
| *300_ieee* | 2382 | 99.89% | 14 | $6.20 \cdot 10^{-1}$ | $\mathbf{6.95 \cdot 10^{-2}}$ | **(8.9)** |
| *500_goc* | 4254 | 99.94% | 14 | $1.81 \cdot 10^{0}$ | $\mathbf{1.40 \cdot 10^{-1}}$ | **(12.9)** |
| *588_sdet* | 4110 | 99.94% | 14 | $1.71 \cdot 10^{0}$ | $\mathbf{1.40 \cdot 10^{-1}}$ | **(12.2)** |
| *793_goc* | 5432 | 99.95% | 14 | $2.96 \cdot 10^{0}$ | $\mathbf{2.65 \cdot 10^{-1}}$ | **(11.2)** |
| *1354_pegase* | 11192 | 99.98% | 18 | $1.58 \cdot 10^{1}$ | $\mathbf{4.14 \cdot 10^{-1}}$ | **(38.1)** |
| *1803_snem* | 15246 | 99.98% | 16 | $3.02 \cdot 10^{1}$ | $\mathbf{7.17 \cdot 10^{-1}}$ | **(42.1)** |
| *1888_rte* | 14480 | 99.98% | 18 | $2.72 \cdot 10^{1}$ | $\mathbf{6.53 \cdot 10^{-1}}$ | **(41.7)** |
| *1951_rte* | 15018 | 99.98% | 20 | $3.10 \cdot 10^{1}$ | $\mathbf{6.50 \cdot 10^{-1}}$ | **(47.7)** |
| *2000_goc* | 19008 | 99.99% | 18 | $6.55 \cdot 10^{1}$ | $\mathbf{1.10 \cdot 10^{0}}$ | **(59.5)** |
| *2312_goc* | 17128 | 99.98% | 16 | $4.43 \cdot 10^{1}$ | $\mathbf{8.69 \cdot 10^{-1}}$ | **(51.0)** |
| *2383wp_k* | 17004 | 99.98% | 16 | $4.39 \cdot 10^{1}$ | $\mathbf{8.48 \cdot 10^{-1}}$ | **(51.7)** |
| *2736sp_k* | 19088 | 99.99% | 14 | $6.31 \cdot 10^{1}$ | $\mathbf{1.02 \cdot 10^{0}}$ | **(62.1)** |
| *2737sop_k* | 18988 | 99.99% | 16 | $5.62 \cdot 10^{1}$ | $\mathbf{1.02 \cdot 10^{0}}$ | **(55.1)** |
| *2742_goc* | 24540 | 99.99% | 14 | $1.37 \cdot 10^{2}$ | $\mathbf{1.11 \cdot 10^{0}}$ | **(122.8)** |
| *2746wop_k* | 19582 | 99.99% | 16 | $6.61 \cdot 10^{1}$ | $\mathbf{1.06 \cdot 10^{0}}$ | **(62.5)** |
| *2746wp_k* | 19520 | 99.99% | 14 | $6.35 \cdot 10^{1}$ | $\mathbf{1.04 \cdot 10^{0}}$ | **(60.9)** |
| *2848_rte* | 21822 | 99.99% | 20 | $8.57 \cdot 10^{1}$ | $\mathbf{1.23 \cdot 10^{0}}$ | **(69.5)** |
| *2853_sdet* | 23028 | 99.99% | 26 | $1.04 \cdot 10^{2}$ | $\mathbf{8.57 \cdot 10^{-1}}$ | **(121.2)** |
| *2868_rte* | 22090 | 99.99% | 20 | $1.10 \cdot 10^{2}$ | $\mathbf{1.27 \cdot 10^{0}}$ | **(86.7)** |
| *2869_pegase* | 25086 | 99.99% | 28 | $1.44 \cdot 10^{2}$ | $\mathbf{1.06 \cdot 10^{0}}$ | **(136.2)** |
| *3012wp_k* | 21082 | 99.99% | 14 | $8.36 \cdot 10^{1}$ | $\mathbf{1.22 \cdot 10^{0}}$ | **(68.4)** |
| *3022_goc* | 23238 | 99.99% | 18 | $1.45 \cdot 10^{2}$ | $\mathbf{9.83 \cdot 10^{-1}}$ | **(147.8)** |
| *3120sp_k* | 21608 | 99.99% | 18 | $9.46 \cdot 10^{1}$ | $\mathbf{1.31 \cdot 10^{0}}$ | **(72.1)** |
| *3375wp_k* | 24350 | 99.99% | 18 | $1.27 \cdot 10^{2}$ | $\mathbf{9.82 \cdot 10^{-1}}$ | **(128.9)** |

[1] Wall time in seconds.
[2] Number of colors resulting from greedy symmetric coloring.
[3] In parentheses: Wall time ratio compared to Symbolics.jl's sparsity detection (higher is better).

Table 5: Performance comparison of Hessian sparsity detection on the Lagrangian of PGLib optimization problems.

| Problem | | Sparsity | | Hessian computation[1] | | | | |
|---|---|---|---|---|---|---|---|---|
| Name | Inputs | Zeros | Colors[2] | AD (prepared) | ASD (prepared)[3] | | ASD (unprepared)[3] | |
| *3_lmbd* | 24 | 91.15% | 6 | $1.82 \cdot 10^{-4}$ | $\mathbf{8.29 \cdot 10^{-5}}$ | **(2.2)** | $1.45 \cdot 10^{-4}$ | (1.3) |
| *5_pjm* | 44 | 94.99% | 8 | $6.33 \cdot 10^{-4}$ | $\mathbf{1.71 \cdot 10^{-4}}$ | **(3.7)** | $3.03 \cdot 10^{-4}$ | (2.1) |
| *14_ieee* | 118 | 97.84% | 10 | $5.38 \cdot 10^{-3}$ | $\mathbf{4.84 \cdot 10^{-4}}$ | **(11.1)** | $1.12 \cdot 10^{-3}$ | (4.8) |
| *24_ieee_rts* | 266 | 99.22% | 12 | $2.56 \cdot 10^{-2}$ | $\mathbf{1.04 \cdot 10^{-3}}$ | **(24.7)** | $2.74 \cdot 10^{-3}$ | (9.3) |
| *30_as* | 236 | 98.89% | 12 | $2.39 \cdot 10^{-2}$ | $\mathbf{1.10 \cdot 10^{-3}}$ | **(21.8)** | $2.84 \cdot 10^{-3}$ | (8.4) |
| *30_ieee* | 236 | 98.89% | 12 | $2.37 \cdot 10^{-2}$ | $\mathbf{1.09 \cdot 10^{-3}}$ | **(21.6)** | $2.87 \cdot 10^{-3}$ | (8.3) |
| *39_epri* | 282 | 99.10% | 10 | $3.28 \cdot 10^{-2}$ | $\mathbf{1.21 \cdot 10^{-3}}$ | **(27.1)** | $3.43 \cdot 10^{-3}$ | (9.6) |
| *57_ieee* | 448 | 99.41% | 14 | $8.80 \cdot 10^{-2}$ | $\mathbf{3.96 \cdot 10^{-3}}$ | **(22.2)** | $9.23 \cdot 10^{-3}$ | (9.5) |
| *60_c* | 518 | 99.56% | 12 | $1.15 \cdot 10^{-1}$ | $\mathbf{2.36 \cdot 10^{-3}}$ | **(48.6)** | $8.61 \cdot 10^{-3}$ | (13.3) |
| *73_ieee_rts* | 824 | 99.74% | 12 | $2.75 \cdot 10^{-1}$ | $\mathbf{3.47 \cdot 10^{-3}}$ | **(79.1)** | $1.54 \cdot 10^{-2}$ | (17.8) |
| *89_pegase* | 1042 | 99.74% | 26 | $5.61 \cdot 10^{-1}$ | $\mathbf{1.61 \cdot 10^{-2}}$ | **(34.8)** | $4.28 \cdot 10^{-2}$ | (13.1) |
| *118_ieee* | 1088 | 99.77% | 12 | $5.55 \cdot 10^{-1}$ | $\mathbf{5.25 \cdot 10^{-3}}$ | **(105.8)** | $3.13 \cdot 10^{-2}$ | (17.7) |
| *162_ieee_dtc* | 1484 | 99.82% | 16 | $1.16 \cdot 10^{0}$ | $\mathbf{1.53 \cdot 10^{-2}}$ | **(75.7)** | $5.53 \cdot 10^{-2}$ | (20.9) |
| *179_goc* | 1468 | 99.83% | 14 | $1.08 \cdot 10^{0}$ | $\mathbf{1.33 \cdot 10^{-2}}$ | **(81.3)** | $5.06 \cdot 10^{-2}$ | (21.4) |
| *197_snem* | 1608 | 99.85% | 14 | $1.34 \cdot 10^{0}$ | $\mathbf{1.46 \cdot 10^{-2}}$ | **(92.2)** | $5.84 \cdot 10^{-2}$ | (23.0) |
| *200_activ* | 1456 | 99.82% | 12 | $1.02 \cdot 10^{0}$ | $\mathbf{6.94 \cdot 10^{-3}}$ | **(146.6)** | $3.88 \cdot 10^{-2}$ | (26.3) |
| *240_pserc* | 2558 | 99.91% | 16 | $3.51 \cdot 10^{0}$ | $\mathbf{2.50 \cdot 10^{-2}}$ | **(140.2)** | $1.04 \cdot 10^{-1}$ | (33.6) |
| *300_ieee* | 2382 | 99.89% | 14 | $3.00 \cdot 10^{0}$ | $\mathbf{2.14 \cdot 10^{-2}}$ | **(140.3)** | $9.67 \cdot 10^{-2}$ | (31.1) |
| *500_goc* | 4254 | 99.94% | 14 | $1.18 \cdot 10^{1}$ | $\mathbf{3.85 \cdot 10^{-2}}$ | **(307.3)** | $2.20 \cdot 10^{-1}$ | (53.7) |
| *588_sdet* | 4110 | 99.94% | 14 | $1.14 \cdot 10^{1}$ | $\mathbf{3.60 \cdot 10^{-2}}$ | **(316.1)** | $2.14 \cdot 10^{-1}$ | (53.3) |
| *793_goc* | 5432 | 99.95% | 14 | $2.17 \cdot 10^{1}$ | $\mathbf{4.91 \cdot 10^{-2}}$ | **(443.1)** | $3.33 \cdot 10^{-1}$ | (65.3) |
| *1354_pegase* | 11192 | 99.98% | 18 | $1.36 \cdot 10^{2}$ | $\mathbf{1.21 \cdot 10^{-1}}$ | **(1128.4)** | $6.21 \cdot 10^{-1}$ | (219.6) |
| *1803_snem* | 15246 | 99.98% | 16 | $2.09 \cdot 10^{2}$ | $\mathbf{1.66 \cdot 10^{-1}}$ | **(1259.5)** | $1.07 \cdot 10^{0}$ | (195.0) |
| *1888_rte* | 14480 | 99.98% | 18 | $8.15 \cdot 10^{2}$ | $\mathbf{1.43 \cdot 10^{-1}}$ | **(5706.7)** | $8.76 \cdot 10^{-1}$ | (930.4) |
| *1951_rte* | 15018 | 99.98% | 20 | $2.00 \cdot 10^{2}$ | $\mathbf{1.54 \cdot 10^{-1}}$ | **(1293.4)** | $1.00 \cdot 10^{0}$ | (199.1) |
| *2000_goc* | 19008 | 99.99% | 18 | $3.58 \cdot 10^{2}$ | $\mathbf{2.15 \cdot 10^{-1}}$ | **(1669.5)** | $1.61 \cdot 10^{0}$ | (222.7) |
| *2312_goc* | 17128 | 99.98% | 16 | $2.75 \cdot 10^{2}$ | $\mathbf{1.87 \cdot 10^{-1}}$ | **(1470.7)** | $1.35 \cdot 10^{0}$ | (204.5) |
| *2383wp_k* | 17004 | 99.98% | 16 | $2.65 \cdot 10^{2}$ | $\mathbf{1.80 \cdot 10^{-1}}$ | **(1468.2)** | $1.14 \cdot 10^{0}$ | (231.4) |
| *2736sp_k* | 19088 | 99.99% | 14 | $3.30 \cdot 10^{2}$ | $\mathbf{1.78 \cdot 10^{-1}}$ | **(1857.2)** | $1.40 \cdot 10^{0}$ | (235.5) |
| *2737sop_k* | 18988 | 99.99% | 16 | $3.29 \cdot 10^{2}$ | $\mathbf{2.02 \cdot 10^{-1}}$ | **(1629.8)** | $1.47 \cdot 10^{0}$ | (223.0) |
| *2742_goc* | 24540 | 99.99% | 14 | $6.50 \cdot 10^{2}$ | $\mathbf{2.41 \cdot 10^{-1}}$ | **(2694.1)** | $1.78 \cdot 10^{0}$ | (366.3) |
| *2746wop_k* | 19582 | 99.99% | 16 | $3.64 \cdot 10^{2}$ | $\mathbf{2.07 \cdot 10^{-1}}$ | **(1755.7)** | $1.54 \cdot 10^{0}$ | (235.6) |
| *2746wp_k* | 19520 | 99.99% | 14 | $3.53 \cdot 10^{2}$ | $\mathbf{1.77 \cdot 10^{-1}}$ | **(1991.4)** | $1.51 \cdot 10^{0}$ | (234.5) |
| *2848_rte* | 21822 | 99.99% | 20 | $4.67 \cdot 10^{2}$ | $\mathbf{2.24 \cdot 10^{-1}}$ | **(2083.5)** | $1.80 \cdot 10^{0}$ | (259.7) |
| *2853_sdet* | 23028 | 99.99% | 26 | $5.38 \cdot 10^{2}$ | $\mathbf{3.62 \cdot 10^{-1}}$ | **(1486.9)** | $1.68 \cdot 10^{0}$ | (320.6) |
| *2868_rte* | 22090 | 99.99% | 20 | $5.02 \cdot 10^{2}$ | $\mathbf{2.35 \cdot 10^{-1}}$ | **(2137.9)** | $1.73 \cdot 10^{0}$ | (290.0) |
| *2869_pegase* | 25086 | 99.99% | 28 | $5.08 \cdot 10^{2}$ | $\mathbf{4.07 \cdot 10^{-1}}$ | **(1249.0)** | $1.99 \cdot 10^{0}$ | (255.5) |
| *3012wp_k* | 21082 | 99.99% | 14 | $4.33 \cdot 10^{2}$ | $\mathbf{1.96 \cdot 10^{-1}}$ | **(2208.3)** | $1.77 \cdot 10^{0}$ | (245.1) |
| *3022_goc* | 23238 | 99.99% | 18 | $5.76 \cdot 10^{2}$ | $\mathbf{2.51 \cdot 10^{-1}}$ | **(2296.9)** | $1.48 \cdot 10^{0}$ | (390.7) |
| *3120sp_k* | 21608 | 99.99% | 18 | $4.56 \cdot 10^{2}$ | $\mathbf{2.26 \cdot 10^{-1}}$ | **(2019.2)** | $1.90 \cdot 10^{0}$ | (240.1) |
| *3375wp_k* | 24350 | 99.99% | 18 | $6.25 \cdot 10^{2}$ | $\mathbf{2.54 \cdot 10^{-1}}$ | **(2463.9)** | $1.71 \cdot 10^{0}$ | (365.1) |

[1] Wall time in seconds.
[2] Number of colors resulting from greedy symmetric coloring.
[3] In parentheses: Wall time ratio compared to prepared AD (higher is better).

Table 6: Performance comparison of AD and ASD Hessian computation on the Lagrangian of PGLib optimization problems.

