# OpenReview forum: "Sparser, Better, Faster, Stronger: Sparsity Detection for Efficient Automatic Differentiation"
_TMLR — Accepted by TMLR_

### Review · Reviewer_vWkC · 2025-02-20

**Summary Of Contributions:**

This paper proposes a new automatic sparse differentiation (ASD) framework for
the Julia programming language.
The ASD framework has three components: a package for propagating
Jacobian/Hessian sparsity patterns during forward passes, a matrix coloring
package for determining the number of necessary vector-Jacobian products,
and an interface for differentiation.
The authors primarily describe how to propagate sparsity patterns, leaving
descriptions of the other two packages to future work.
During the forward pass, sparsity patterns are computed from parent nodes
in the compute graph using local propagation rules, much like the "seed" in
forward mode differentiation.
The output value of the forward procedure gives the sparsity pattern of the
Jacobian/Hessian and can be used for matrix coloring and compressed
differentiation.
Numerical simulations show that this approach is much faster than previous
software for automatic sparsity detection in Julia.

**Audience:**

Yes

**Broader Impact Concerns:**

None.

**Claims And Evidence:**

Yes

**Requested Changes:**

**Questions**:

- What is the motivation for presenting `SparseMatrixColorings.jl` and
  `DifferentiationInterface.jl` in separate (future) papers?
- Can SCT significantly improve training time for ML problems with sparse
  Jacobians, such as training ReLU networks?
- Was significant code/performance optimization necessary for SCT to be fast?
- What were the main challenges in the implementation and why are they
  interesting to the TMLR community?

**Strengths And Weaknesses:**

**Strengths**:

- The SCT sparsity detector proposed by the authors is two to three orders of
  magnitude faster on benchmark problems than previous software in Julia.

- Experiments show that SCT reduces to the overhead of ASD, making it faster
  than idealized dense automatic differentiation.

- The description of forward/reverse mode differentiation and ASD is clear.


**Weaknesses**:

- The paper describes only the first part of the ASD pipeline, leaving matrix
  coloring and the differentiation interface to "future papers".

- No experiments are given for machine learning (ML) applications.

- The conceptual novelty and implementation difficulty for the authors ASD
  approach is not clear.


### Additional Details

This is an interesting paper which outlines a new software package for
leveraging sparsity in automatic differentiation.
The paper is well-written and provides clear general description the
ASD package and its theoretical principles.
The package itself is written in Julia, a widely used language in the optimization and
scientific computing communities.
While the experimental evaluation is rather small in scope, it shows that SCT
outperforms existing software by several orders of magnitude.
Taking these items in to account, I think the package is of sufficient interest
to the TMLR community.
Unfortunately, the positive aspects of the paper are outweighed by my concerns
regrading the paper's weaknesses.
These issues, which are discussed in detail below, must be reasonably addressed
before I can support accepting this paper.

**Small Scope**:
This paper only describes the SCT sparsity detector and explicitly leaves the
rest of the ASD pipeline to future work.
This has three effects: (i) it makes it difficult to weigh the contributions
of the sparsity detector compared to the other elements of the pipeline; (ii)
it makes it seem like the authors are trying to "slice their contributions
thin" and maximize the number of publications obtained from this work; and
(iii) it harms the pedagogical value of the paper as the reader learns nothing
about matrix coloring or the differentiation interface.
I strongly urge the authors to consider revising their paper to include
descriptions of all three software components.
These could easily be fit into the paper by paring down on the five pages of
introduction/related work and compressing verbose parts of the text.

**ML Experiments**:
If the authors only wish to describe SCT, then I believe it is critical that
clear applications of the package are developed for ML problems.
As the authors note, automatic differentiation is a key tool in ML and many ML
problems exhibit Jacobian/Hessian sparsity.
Experiments showing that faster ASD has a positive down-stream affect on ML
tasks would make an excellent argument for the utility of the software package.
It would also increase the papers contributions, which I consider to be quite
thin.
One obvious problem the authors could consider is training ReLU networks; as
the text notes, ReLU activations induce Jacobian sparsity, so perhaps ASD would
be useful here.

**Contribution Size**:
Automatic differentiation is not my area of expertise and I found it hard to
quantify the novelty/interest of the software using the text.
The authors comment that they present a "revised viewpoint" on sparsity
detection, but this viewpoint seems quite standard and intuitive to me given
familiarity with forward-mode differentiation.
Similarly, the authors make no comment on the actual implementation of the package.
Was significant code/performance optimization necessary for SCT to be fast?
What were the main challenges in the implementation and why are they interesting?
Answering these questions would clarify the core contributions of the paper.

### Minor Issues:

- Page 1: "...resting assured that gradients will be computed efficiently and
  correctly without human intervention." --- Actually, there are several
  well-known bugs in automatic differentiation systems as result of iterative
  application of sub-gradient computation rules. For example, functions like
  $\min\{x, \max\{x, 0\}\} = x$ are often differentiated incorrectly in PyTorch.

- Page 2: "...most advanced automatic sparse differentiation system" --- what
  do you mean by "most advanced"? Does this mean fastest, best at detecting
  sparsity, etc? I think this statement should be made based on clear metrics.

- Page 5: "Specifically for Hessian patterns, Walther (2008) extends the
  operator overloading approach..." --- it is more typical to use "they" as a
  genderless singular pronoun when referring to papers with only one author.

- Eq. (1): Equations are part of a sentence like any other textual element.
  This equation stands alone as its own sentence, which is slightly odd.

- Page 7: "These binary values us whether the" --- missing word?

- Perhaps the authors should consider submitting this work to a more
  software-focused venue, such as JMLR's [machine learning open source
  software](https://jmlr.org/mloss/mloss-info.html) track.  While I don't think
  this paper is necessarily a bad fit for TMLR, it may receive more attention
  at software-focused venues.

---

> ### Author Response · Authors · 2025-03-28
>
> We are grateful to the reviewer for their suggestions and remarks, and are happy to see that our improvements against the state of the art are recognized.
>
> ### Weaknesses
>
> > The paper describes only the first part of the ASD pipeline, leaving matrix coloring and the differentiation interface to "future papers".
>
> We understand this limitation and explain it in our global response.
>
> > No experiments are given for machine learning (ML) applications.
>
> While we understand that it is not standard deep learning, ODEs and optimization problems like the ACOPF are very relevant in scientific ML, where physical simulations are inserted into differentiable models.
> We have tried to make this more explicit by adding relevant citations, e.g. on universal differential equations.
>
> We have also highlighted the experiments on CNN (appendix E.1) and added a new experiment on GNN and implicit differentiation (appendix G).
>
> > The conceptual novelty and implementation difficulty for the authors ASD approach is not clear.
>
> We have strived to make the novelties more explicit in the contributions (section 1.3), and completely rewritten Sec. 4 to highlight implementation and algorithmic challenges we tackled.
>
> ### Additional Details
>
> > **Small Scope**: This paper only describes the SCT sparsity detector and explicitly leaves the rest of the ASD pipeline to future work. [...] I strongly urge the authors to consider revising their paper to include descriptions of all three software components. These could easily be fit into the paper by paring down on the five pages of introduction/related work and compressing verbose parts of the text.
>
> While we understand that the introduction may seem verbose, part of the goal for this paper is to act as a tutorial for ML researchers unfamiliar with ASD.
> Therefore, we believe our exposition of related work has intrinsic value for the community.
>
> As for the lack of description of the remaining pipeline, we discuss this in our global response.
>
> > **ML Experiments**: If the authors only wish to describe SCT, then I believe it is critical that clear applications of the package are developed for ML problems. [...] One obvious problem the authors could consider is training ReLU networks [...]
>
> Unfortunately, ASD cannot help train ReLU networks because computing a gradient requires a single reverse pass through the network.
> However, sparsity detection can help analyze the network a posteriori to detect dependencies between inputs and outputs (as shown in appendix E.1).
> Furthermore, Jacobians can be useful whenever linear systems must be solved during training, as is the case for implicit networks.
> We have added a new experiment for implicit differentiation of GNNs in appendix G.
> Other potential applications to ML are listed in Sec. 1.2 and appendix A.
>
> > **Contribution Size**: Automatic differentiation is not my area of expertise and I found it hard to quantify the novelty/interest of the software using the text.
>
> We have made our contributions more explicit. On the methodological side, the flexible index set representations with multiple dispatch, the classification of operators based on derivative types and the tensor overloads are additions/perspectives which do not seem well-known in the literature.
>
> As for the implementation, we rewrote Sec. 4 in order to discuss the main features of SCT, its interactions with downstream packages, its generality as well as limitations.
>
> **Minor Issues**
>
> > Page 1: "...resting assured that gradients will be computed efficiently and correctly without human intervention." --- Actually, there are several well-known bugs in automatic differentiation [...]
>
> We added a footnote acknowledging AD's pitfalls.
>
> > Page 2: "...most advanced automatic sparse differentiation system" --- what do you mean by "most advanced"? [...]
>
> We removed this statement and instead focus on factual results from benchmarks.
>
> > Perhaps the authors should consider submitting this work to a more software-focused venue [...]
>
> We appreciate this recommendation.
> However, all reviewers agreed that this paper was interesting for the TMLR audience, so we hope its value will be recognized here.
>
> We also fixed the genderless pronoun, stand-alone equation and missing word.
>
> ### Requested Changes
>
> Questions:
>
> > What is the motivation for presenting SparseMatrixColorings.jl and DifferentiationInterface.jl in separate (future) papers?
>
> We answer this question in our global response.
>
> > Can SCT significantly improve training time for ML problems with sparse Jacobians, such as training ReLU networks?
>
> See our response above.
>
> > Was significant code/performance optimization necessary for SCT to be fast? What were the main challenges in the implementation and why are they interesting to the TMLR community?
>
> We added more details in Sec. 4, and present new experiments on ML applications in appendix G.

---

> > ### Comment · Reviewer_vWkC · 2025-04-01
> >
> > Thanks for your response and for posting the revision. Here are my replies.
> >
> > > The literature on AD contains many papers that focus solely on either sparsity detection or coloring [and] sparsity detection is of independent interest.
> >
> > This may be true, but this is a software paper primarily arguing that the machine learning community should adopt your ASD pipeline. That pipeline includes the sparsity detection package and as well as the new sparse metric coloring and differentiation interface packages. The ASD experiments in the paper also make use of all three packages. I would more amenable to this line of argument if the sparsity detection package had also been integrated into existing ASD frameworks, thus demonstrating its utility as a stand-alone contribution. I expect such an approach is possible given the modularity of your approach.
> >
> > > The first author of the detection paper has no involvement in coloring and vice-versa, because both projects were developed independently.
> >
> > While I appreciate these project management difficulties, they don't affect my evaluation of the paper. I still think it is greatly preferable from a research standpoint to merge the projects to have one strong paper with a full description of the ASD pipeline rather than two or more weak papers.
> >
> > > We have strived to make the novelties more explicit in the contributions (section 1.3), and completely rewritten Sec. 4 to highlight implementation and algorithmic challenges we tackled.
> >
> > I like the rewrite of Section 4. Expanding on the implementation details clarifies your contributions and strengthens the paper. Regarding Section 4.1, is there any way of automatically detecting whether bit vectors or hash tables will be more performant, perhaps based on the available hardware and the problem dimension? It seems like this would be greatly preferred over having the user determine which to use on a case-by-case basis.
> >
> > > Unfortunately, ASD cannot help train ReLU networks because computing a gradient requires a single reverse pass through the network.
> >
> > I was thinking about training algorithms that require computing the entire Jacobian (sometimes called "individual gradients") rather than the computing the vector-Jacobian product. For example, see the many papers cited by Dangel et al. [1]. For a wide two-layer ReLU network, I would expect this Jacobian to be quite sparse and thus GGN-type methods might benefit ASD.
> >
> > > We have also highlighted the experiments on CNN (appendix E.1) and added a new experiment on GNN and implicit differentiation (appendix G).
> >
> > These are positive changes and I like the experiments with GNNs. I agree with Reviewer PDA9 that they would be best if presented in the main paper. However, I don't like that Section 5 now omits that  `SparseMatrixColorings.jl` and `DifferentiationInterface.jl` are also written by the authors; this seems like poor way to side-step the contribution issue. However, it does fix the fact that the previous wording came quite close to violating double blind. Next time I suggest using anonymized references for these packages.
> >
> > [1] Dangel, Felix, Frederik Kunstner, and Philipp Hennig. "BackPACK: Packing more into Backprop." International Conference on Learning Representations (ICLR). 2020.

---

### Review · Reviewer_BWxp · 2025-02-24

**Summary Of Contributions:**

This paper deals with Automatic Sparse Differentiation (ASD), a set of techniques that leverage the sparsity of Jacobian/Hessian to speed up their computation. The paper presents propagation rules for sparsity pattern detection to approximate the sparsity pattern of Jacobian and Hessian matrices, thanks to a binarization of the chain rule. These propagation rules are the cornerstone of efficient sparsity detection and are implemented in a Julia package, as well as the matrix coloring and differentiation steps. The sparsity detection implementation for Jacobian and Hessian is benchmarked against another sparsity detection technique. Moreover, the full ASD pipeline is numerically compared with AD.

**Audience:**

Yes

**Claims And Evidence:**

Yes

**Requested Changes:**

### Minor
* **p.9 and p.10**: Since the reference (Chen et al., 2018) is straddling on pages 9 and 10, the hyperlink to the reference covers the footnote of page 9 and the header of page 10. A solution is proposed [here](https://tex.stackexchange.com/questions/330669/cite-hyperlink-across-two-pages).


### Clarity
* **p.6 and p.7**: The unfamiliar reader can be confused by the appearance of $y$, without being defined. The author should clarify that $y(x)$ is an intermediate value in the computation of $f_i(x)$. Moreover, I don't get why when it is written "$y\in\mathbb{R}$" there is no argument $x$ and when it is written "$1[\nabla y(x)]\in\{0,1\}^n$ the argument is written.

* **Notation confusion in sparcity pattern**: I wonder if there is confusion in the notation between the actual sparsity pattern $1[\nabla \gamma]$ (as defined in the notation section) and the "approximate" sparsity pattern contained in the tracer. Indeed, if I understand correctly, the equation (2) tells us that $1[\nabla \gamma] = 0$ if and only if $1[\partial_1 \varphi]\cdot 1[\nabla \alpha] = 1[\partial_2 \varphi]\cdot 1[\nabla \beta] = 0$. This is not true if $\varphi(\alpha, \beta) = \alpha + \beta$ and $\alpha(x) = x = -\beta(x)$. In this case, for $x\neq 0$ we have $1[\nabla \gamma(x)] = 0$ and  $1[\partial_1 \varphi]\cdot 1[\nabla \alpha] = 1[\partial_2 \varphi]\cdot 1[\nabla \beta] = 1$. What is true is that
$
1[\nabla \gamma] \leq 1[\partial_1\varphi]\cdot 1[\nabla \alpha] \vee 1[\partial_2\varphi]\cdot 1[\nabla \beta]
$. This tells us that the propagation rule in the right-hand-side is not exact but it does not approximate by 0 a non-zero coefficient of $1[\nabla \gamma]$, which is crucial.
Thus, I think that in equations (2) and (3) (for the Hessian), there should either be a $\leq$ instead of $=$ or the notation should be changed to reflect that the sparsity pattern is not exact but an approximation.

**Strengths And Weaknesses:**

### Strengths
* This paper brings Automatic Sparse Differentiation to the machine learning community. It can be convenient for problems that require the derivation of sparse Jacobian/Hessian matrices and for which computing some Jacobian/Hessian-vector products is not sufficient.

* The authors propose an efficient implementation of the ASD pipeline in Julia.

* An extensive benchmark of ASD against AD is provided, showing the efficiency of ASD in practice.

* The paper is very well written and the presentation is clear.


### Weaknesses
* See the requested changes section for minor issues.

---

> ### Author Response · Authors · 2025-03-28
>
> We thank the reviewer for their comments, and for acknowledging that our paper brings sparse differentiation to the ML community in an efficient manner.
>
> ### Weaknesses
>
> > See the requested changes section for minor issues.
>
> ### Requested Changes
>
> **Minor**
>
> > p.9 and p.10: Since the reference (Chen et al., 2018) is straddling on pages 9 and 10, the hyperlink to the reference covers the footnote of page 9 and the header of page 10. A solution is proposed here.
>
> The experimental section was rewritten and the problem seems to be gone.
>
> **Clarity**
>
> > p.6 and p.7: The unfamiliar reader can be confused by the appearance of $y$, without being defined. The author should clarify that is an intermediate value in the computation of . Moreover, I don't get why when it is written "$y \in \mathbb{R}$" there is no argument and when it is written $1[\nabla y(x)] \in \{0, 1\}$" the argument is written.
>
> We have clarified what $y(x)$ stands for and fixed the notation.
>
> > Notation confusion in sparsity pattern: I wonder if there is confusion in the notation between the actual sparsity pattern $1[\nabla \gamma]$ (as defined in the notation section) and the "approximate" sparsity pattern contained in the tracer. Indeed, if I understand correctly, the equation (2) tells us that $1[\nabla \gamma] = 0$ if and only if $1[\partial_1 \varphi] \cdot 1[\nabla \alpha] = 1[\partial_2 \varphi] \cdot 1[\nabla \beta] = 0$. This is not true if $\varphi(\alpha, \beta) = \alpha + \beta$ and $\alpha(x) = x = -\beta(x)$. In this case, for $x \neq 0$ we have $1[\nabla \gamma(x)] = 0$ and $1[\partial_1 \varphi] \cdot 1[\nabla \alpha] = 1[\partial_2 \varphi] \cdot 1[\nabla \beta] = 0$. What is true is that $1[\nabla \gamma] \leq 1[\partial_1 \varphi] \cdot 1[\nabla \alpha] \vee 1[\partial_2 \varphi] \cdot 1[\nabla \beta]$. This tells us that the propagation rule in the right-hand-side is not exact but it does not approximate by $0$ a non-zero coefficient of $1[\nabla \gamma]$, which is crucial. Thus, I think that in equations (2) and (3) (for the Hessian), there should either be a instead of or the notation should be changed to reflect that the sparsity pattern is not exact but an approximation.
>
> This is an important and accurate remark: the sparsity patterns we estimate are overly conservative, and cannot account for accidental cancellations of this type.
>
> We have accounted for this by replacing the equal signs with $\leq$ signs, to highlight the overestimation.
> This crucial detail is discussed in remark 2.

---

### Review · Reviewer_ksEs · 2025-03-19

**Summary Of Contributions:**

The paper proposes a new exposition and implementation of the Automatic Sparse Differentiation (ASD) based on the operator overloading concept. In detail, the authors argue that there exists a certain barrier between two scientific communities: the community developing Automatic Differentiation (AD) algorithms and the Machine Learning (ML) community. In particular, Automatic Sparse Differentiation is still not widely adopted by the ML community despite being a very promising direction for optimizing the efficiency of ML algorithms.

In order to bridge the gap between the communities and make ASD more practical, the authors provide a minimalistic description of the operator overloading approach and unify several previous ASD methods in a single Julia library. They provide an open-source implementation of the library and benchmark it on several tasks relevant to the ML community. The benchmarks demonstrate the great utility of ASD with operator overloading in terms of the wall time required for the Jacobian and Hessian computations.

**Audience:**

Yes

**Broader Impact Concerns:**

The paper does not raise any broader impact concerns.

**Claims And Evidence:**

Yes

**Requested Changes:**

The major change I request is the addition of verbatim code snippets demonstrating the code difference between AD and ASD. This would greatly improve readability and help achieve the objective of promoting advanced AD techniques.

The list of minor comments:
- abstract, page 3, the following phrase requires an explanation "one-off computations of Jacobians"
- page 4, second to the last paragraph, typo in "to a select few AD systems";
- page 6, the usage of color in eq. (1) and the one before is not clear, i.e. what exactly the colour show here and why is the colour not used in other equations?;
- page 8, first line, typo in "$x_j$ into $x_j$ into";
- page 9, last paragraph of Section 4, typo in "Not only is it more generic with respect to the AD backend, it also much faster" and "An demonstration of the code";
- Figure 2, the resolution of the presented sparsity patterns is very low, in stark contrast with Figure 1;
- page 11, the following fact is not evident from Table 3 as suggested "sparsity detection alone took more wall time than the full AD Jacobian computation";
- Tables 3 and 4, it is not clear what "pattern detection" stands for. I would suggest being consistent with the text where the authors use "sparsity detection";
- why does Table 3 have only pattern detection wall time and does not have a full computation wall time comparison?

**Strengths And Weaknesses:**

As the authors mention in the paper, the Automatic Differentiation and Machine Learning communities are quite separated. Therefore, it must be noted that my review is provided from the perspective of a member of the ML community. This implies that I might not be familiar with some methods developed by the AD community, and I approach these algorithms from the perspective of a user.

Strengths:
- The paper provides a clear exposition of the previous works and the proposed method. It is immediately clear where one can expect the acceleration of the computation and why.
- The method demonstrates very strong empirical results on the selected tasks. That is, there is no question that the proposed method improves the efficiency of the computation and will do so for different hardware, problems, and hyperparameters. This simplifies comparison and serves as a good motivation for the usage of the proposed method.
- The authors supplement an open-source implementation of the proposed method as a single Julia library. This is essential for achieving the declared goals of disseminating advanced AD techniques among the ML community.

Weaknesses:
- The Julia implementation is not the best or the fastest way to spread your method among the ML community. Although I don't have a clear understanding of this design choice, this is a clear barrier that will prevent many practitioners from using the proposed method.
- The CPU-based implementation does not help achieve the declared goals. Indeed, as the authors mention in their work, CPU-based solutions are currently not the main focus of the ML community. Furthermore, the parallel nature of computations on GPU raises concerns about the possibility of implementing the proposed algorithm on GPU and the potential benefits from ASD.
- The paper lacks code examples explaining to the reader how to use the proposed framework. Since the goal of the paper is, in part, educational, it would be more convincing if the authors demonstrated how one can easily use the proposed method instead of conventional AD methods.

---

> ### Author Response · Authors · 2025-03-28
>
> We are grateful to the reviewer for their thorough evaluation, and for highlighting our clear exposition and strong empirical results.
>
> ### Weaknesses
>
> > The Julia implementation is not the best or the fastest way to spread your method among the ML community. Although I don't have a clear understanding of this design choice, this is a clear barrier that will prevent many practitioners from using the proposed method.
>
> We acknowledge that this is a barrier to wider adoption, and explain the reasons for our choice in our global response.
>
> > The CPU-based implementation does not help achieve the declared goals. Indeed, as the authors mention in their work, CPU-based solutions are currently not the main focus of the ML community. Furthermore, the parallel nature of computations on GPU raises concerns about the possibility of implementing the proposed algorithm on GPU and the potential benefits from ASD.
>
> It is true that our CPU implementation is not appropriate when working on large-scale neural networks.
> However, it is well-suited to medium-scale scientific ML problems such as those described above, and those benchmarked in our paper.
> We discuss this more in our global response.
>
> > The paper lacks code examples explaining to the reader how to use the proposed framework. Since the goal of the paper is, in part, educational, it would be more convincing if the authors demonstrated how one can easily use the proposed method instead of conventional AD methods.
>
> We have highlighted code examples and demonstrations in appendix E.
>
> ### Requested Changes
>
> > The major change I request is the addition of verbatim code snippets demonstrating the code difference between AD and ASD. This would greatly improve readability and help achieve the objective of promoting advanced AD techniques.
>
> These examples are located in appendix E, since the 12-page limit did not allow putting them in the main body of the paper.
> More examples are available in the documentation of the respective packages, e.g. <https://juliadiff.org/DifferentiationInterface.jl/DifferentiationInterface/stable/tutorials/advanced/#Sparsity>.
>
> **The list of minor comments**
>
> > abstract, page 3, the following phrase requires an explanation "one-off computations of Jacobians"
>
> We added an explanation that "one-off" refers to the absence of amortization for the preparation step.
>
> > page 4, second to the last paragraph, typo in "to a select few AD systems";
>
> We double-checked and think this is actually correct English.
>
> > page 6, the usage of color in eq. (1) and the one before is not clear, i.e. what exactly the colour show here and why is the colour not used in other equations?;
>
> We removed the color from the derivation of first-order equations.
> It is mostly useful in appendix C, to keep track of terms inside second-order derivations.
>
> > page 8, first line, typo in "$x_j$ into $x_j$ into"
>
> We corrected this.
>
> > page 9, last paragraph of Section 4, typo in "Not only is it more generic with respect to the AD backend, it also much faster" and "An demonstration of the code";
>
> We completely rewrote section 4.
>
> > Figure 2, the resolution of the presented sparsity patterns is very low, in stark contrast with Figure 1;
>
> We zoomed in on the PDF and found the resolution to be satisfactory, can the reviewer explain what they  meant?
>
> > page 11, the following fact is not evident from Table 3 as suggested "sparsity detection alone took more wall time than the full AD Jacobian computation";
>
> We added Figure 2 where the benchmark times are decomposed into sparsity detection, coloring and actual differentiation with JVPs. This should help make such comparisons clearer.
>
> > Tables 3 and 4, it is not clear what "pattern detection" stands for. I would suggest being consistent with the text where the authors use "sparsity detection";
>
> We made the terminology more consistent across the paper.
>
> > why does Table 3 have only pattern detection wall time and does not have a full computation wall time comparison?
>
> We chose to split these tables for space reasons, and because our main focus is the sparsity detection aspect.

---

### Review · Reviewer_PDA9 · 2025-03-19

**Summary Of Contributions:**

The paper presents a new pipeline to compute efficiently sparse Jacobians or Hessians in a differentiable programming framework. To be exact, the underlying algorithms are not new but the package itself, coded in Julia, is new, versatile and demonstrates improvements compared to existing libraries in Julia.
The authors explain in detail the benefits and applications of automatic differentiation libraries able to take advantage of sparse Jacobians/Hessians. They also provide a detailed related work of this old subject.
The authors focus, in this manuscript, on detecting sparsity patterns in the Jacobian and give a new presentation on the subject based on operator overloading. A second essential step (matrix coloring) is relegated to another manuscript.
Experiments demonstrate that the new package outperforms previous packages by an order of magnitude in speed in terms of sparsity pattern detection. Experiments also demonstrate that the new package is competitive at computing the whole Jacobian, even without considering amortizing the cost of sparsity detection over several Jacobian computations.

**Audience:**

Yes

**Claims And Evidence:**

Yes

**Requested Changes:**

Main changes:
- I believe the manuscript would benefit from being fused with the ones that would describe the matrix coloring and the overall automatic differentiation interface. Otherwise the authors will end up with three incomplete papers with clear overlap of contributions and even maybe content. As much as I appreciate the current manuscript, I think such practice (dividing papers in smaller papers), may not be well regarded by the community.
- Add experiments such as:
  - Speed to make one optimization step in the ACOPF problem. More generally, comparison between "lazy products" and the proposed package to solve linear system like $Ha =b$ or $Ja = b$. I suppose the ACOPF problem solves such a linear system every step so it can be a starter. In optimization, the sparse Jacobians could particularly shine to solve nonlinear least-square problems where iterative algorithms use linear systems defined typically by the Jacobian.
  - Comparison with closed source package if possible.

Clarification questions:
- About the prepared ASD case, since the goal is to compute just one Jacobian/Hessian matrix how can the costs be amortized? For global sparsity patterns, it is clear that the global patterns can be reused to compute sparse Jacobians/Hessians at other points. But the local sparsity patterns, as presented by the authors, will depend each time on the point at which the Jacobian/Hessian is computed and cannot be amortized no?
- "batched evaluation" refers to vmap in jax?


Details
- There has also been a recent efforts about optimizing automatic differentiation engines through reinforcement learning (using vertex elimination and alpha zero) [1]. Comparing the present library is probably out of scope but worth mentioning nevertheless
- The authors may consider avoiding phrasing that do not age well like "most advanced automatic sparse differentiation system" (just remove "most"). Similarly do not refer to "future papers"
- page 3 top, "Python currently lack such tooling" what tooling exactly? Automatic sparse differentiation? or multiple dispatch paradygm?
- Spectral analyses of Hessians or Jacobians could also largely benefit from having access to the actual matrix. In deep learning, there has been a growing interest in understanding the spectral properties of the Hessian of the objective along training for example [2]. It could be worth mentioning it.
- "It is worth mentioning that some optimization solvers only accept explicit Jacobians/Hessian matrices" -> give references. Also is it a bottleneck in terms of code or algorithm?
- "optimization solvers often welcome matrices in CSC or CSR format" -> give reference
- While finding local sparsity patterns is clearly explained through operator overloading, finding global sparsity patterns is not really explained.
- "This allows us to bypass the original scalar computational graph... " -> this sentence is not clear. More importantly the scope of the tensor-overloads part is not clear. It would be important for the reader to understand whether future work is needed to optimize the approach for convolutions.
- Just before section 5: "An demonstration" -> "A demonstration"
- Appendix A.5 "In the end, the right choice of set implementation will depend on ..." What set implementations the authors are talking about? Is it referring to various sparse encoding formats?

Suggestions (feel free to ignore):
- May consider a table summarizing the different approaches for automatic sparse differentiation and their main differences if there are.
- Maybe mention in the notations that the operator $\cdot$ has priority over the operator $\vee$ in the diverse mathematical formulations.
- The sentence "We can apply the previous remarks on ..." is somewhat weirdly formulated. Could be replaced by two sentences "The gradient $g = \nabla f(x)$ can be computed by reverse mode AD. Forward mode AD can then be applied on the computation of $g$"
- May put the coding block (page 23) in main text, since this manuscript is presenting the library.

[1] Optimizing Automatic Differentiation with Deep Reinforcement Learning, Lohoff et al, NeurIPS 2024
[2] Gradient Descent on Neural Networks Typically Occurs at the Edge of Stability, Cohen et al, ICLR 2021

**Strengths And Weaknesses:**

Strengths:
- The paper is very well presented with a great description of earlier work.
- The presentation of sparsity detection patterns with operator overloading is very clear.
- The experiments demonstrate the benefits of the approach in Julia

Weaknesses:
- The whole pipeline is not presented. In my opinion, the matrix coloring part should be presented to give a full overview. Since the contribution of the paper is the pipeline and not a new algorithm, an incomplete pipeline makes the paper simply incomplete.
- Experiments can be made more extensive. As the authors mention at the beginning, "matrix-free" methods have been largely adopted to avoid full Jacobian/Hessian computations. It would be great to see the performance of the proposed package in downstream tasks like solving a linear system. By having access to the whole Jacobian, the authors may benefit from methods that are more numerically stable for example. It's unclear though it the cost of detecting the sparsity pattern could be amortized in this regime. Full experimental details on such questions would largely benefit the manuscript.

---

> ### Author Response · Authors · 2025-03-28
>
> We thank the reviewer for their time and expertise, and for underlining the quality of our presentation and experiments.
>
> ### Weaknesses
>
> > The whole pipeline is not presented. In my opinion, the matrix coloring part should be presented to give a full overview. Since the contribution of the paper is the pipeline and not a new algorithm, an incomplete pipeline makes the paper simply incomplete.
>
> We understand this criticism, and address it in our global response.
>
> > Experiments can be made more extensive. As the authors mention at the beginning, "matrix-free" methods have been largely adopted to avoid full Jacobian/Hessian computations. It would be great to see the performance of the proposed package in downstream tasks like solving a linear system. By having access to the whole Jacobian, the authors may benefit from methods that are more numerically stable for example. It's unclear though it the cost of detecting the sparsity pattern could be amortized in this regime. Full experimental details on such questions would largely benefit the manuscript.
>
> We thank the reviewer for this suggestion, which we implemented in the revision. Section 2.2 discusses iterative and direct solvers more at length, while experimental results are presented in appendices F.2 and G.
>
> ### Requested Changes
>
> **Main changes**
>
> > I believe the manuscript would benefit from being fused with the ones that would describe the matrix coloring and the overall automatic differentiation interface.
>
> We appreciate the suggestion, but as argued above, we strongly feel that these aspects deserve independent publications.
>
> > Add experiments such as:
> > - Speed to make one optimization step in the ACOPF problem. More generally, comparison between "lazy products" and the proposed package to solve linear system like $Ha = b$ or $Ja = b$. [...] In optimization, the sparse Jacobians could particularly shine to solve nonlinear least-square problems where iterative algorithms use linear systems defined typically by the Jacobian.
>
> We added experiments on linear systems in appendices F.2 and G that are focused on Jacobian matrices for the Brusselator and for implicit GNNs.
> In the case of the ACOPF, non-convexity means that the Hessian is not positive-definite, which would have required further approximations or convexifications to implement a proper Newton step. We thus left it for future work.
>
> > - Comparison with closed source package if possible.
>
> Comparison with closed source is not possible because sparsity detection is language specific, and thus our method only applies to functions coded in Julia (where we compared to the state-of-the-art package).
>
> **Clarification questions**
>
> > About the prepared ASD case, since the goal is to compute just one Jacobian/Hessian matrix how can the costs be amortized? For global sparsity patterns, it is clear that the global patterns can be reused to compute sparse Jacobians/Hessians at other points. But the local sparsity patterns, as presented by the authors, will depend each time on the point at which the Jacobian/Hessian is computed and cannot be amortized no?
>
> Indeed, while global sparsity patterns are reusable by design, local ones generally depend on the input.
> At first glance, this means that local patterns cannot be amortized.
> However, we also show in our benchmarks that sparsity detection doesn't always need amortization to be useful.
>
> Furthermore, in some cases, local sparsity patterns can be used to circumvent "useless" control flow.
> For instance, a statement like the following does not make sense for global tracers because they do not have a primal value:
>
> ```julia
> if x < y
>     return x
> else
>     error()
> end
> ```
>
> Running this code will error if `x` or `y` are global tracers, but it will work fine with local tracers.
> This is a fundamental limitation of operator overloading, which could be overcome with some amount of source transformation.
> In the meantime, we have observed that users are often able to use local tracers as proxies to handle these cases, whenever control flow is not expected to change the sparsity pattern.
>
> Finally, as our Figure 4b demonstrates, local sparsity detection leads to more sparsity than global sparsity detection, which means it could be faster in theory for some scenarios.
>
> > "batched evaluation" refers to vmap in jax?
>
> In theory yes, but in practice Julia AD backends only support batches of limited size $b \approx 16$ to optimize memory bandwidth.
> The typical way to compute Jacobians or Hessians is therefore to split the input dimension $n$ into $n/b$ batches of size $b$.

---

> > ### Author Response · Authors · 2025-03-28
> >
> > **Details**
> >
> > > There has also been a recent efforts about optimizing automatic differentiation engines through reinforcement learning [...]
> >
> > Thank you for this very relevant pointer, which we have added to our review.
> >
> > > The authors may consider avoiding phrasing that do not age well [...]
> >
> > We have removed these formulations.
> >
> > > page 3 top, "Python currently lack such tooling" what tooling exactly? Automatic sparse differentiation? or multiple dispatch paradygm?
> >
> > ASD in general is practically non-existent in Python, except through domain-specific languages like CasADI, and in the recent Graphax.
> > This is discussed in our survey of high-level implementations (appendix B), and we have added a footnote pointing readers to it.
> >
> > > Spectral analyses of Hessians or Jacobians could also largely benefit from having access to the actual matrix. In deep learning, there has been a growing interest in understanding the spectral properties of the Hessian of the objective along training for example [2].
> >
> > We have added this to our motivation paragraph.
> >
> > > "It is worth mentioning that some optimization solvers only accept explicit Jacobians/Hessian matrices" -> give references. [...]
> > > "optimization solvers often welcome matrices in CSC or CSR format" -> give reference
> >
> > We have added mentions of two specific solvers, Ipopt and Knitro.
> >
> > The bottleneck is both in terms of code and algorithm.
> > In general, direct solvers are less vulnerable to ill-conditioning, which occurs frequently in the KKT system of interior point methods, and therefore more generic (no need to come up with a custom preconditioner).
> > That is probably part of the reason why many solvers only accept materialized matrices when linear systems must be solved.
> >
> > > While finding local sparsity patterns is clearly explained through operator overloading, finding global sparsity patterns is not really explained.
> >
> > Global sparsity detection works in exactly the same way, but removing the value stored inside the tracer.
> > This was indeed unclear, and we have added more explanations to section 4, especially in 4.3.
> >
> > > [...] the scope of the tensor-overloads part is not clear. It would be important for the reader to understand whether future work is needed to optimize the approach for convolutions.
> >
> > Further work is definitely needed on array-level overloads.
> > The one we present is a proof of concept showing that it can be achieved, but functions like the convolution are more sophisticated and their implementation requires handling a lot of edge cases.
> > If we are okay with overly conservative sparsity patterns, we can make bolder approximations and take more shortcuts.
> >
> > We have added more details on array overloads to section 4.2.
> >
> > > Appendix A.5 "In the end, the right choice of set implementation will depend on ..." What set implementations the authors are talking about?
> >
> > We have extended this discussion of set encodings in section 4.1.
> >
> > **Suggestions (feel free to ignore)**
> >
> > > mention in the notations that the operator $\cdot$ has priority over the operator $\vee$
> >
> > We have mentioned this.
> >
> > > The sentence "We can apply the previous remarks on ..." is somewhat weirdly formulated.
> >
> > We have rephrased this.
> >
> > > May put the coding block (page 23) in main text, since this manuscript is presenting the library.
> >
> > For lack of space we kept the code demonstrations in the appendix.
> > This is also coherent with our increased focus on sparsity detection, highlighting that the broader pipeline is not the topic of this paper.

---

> > > ### Comment · Reviewer_PDA9 · 2025-03-28
> > > **Thank you for the revision!**
> > >
> > > Thank you for your answers and for the thorough pass on your manuscript.
> > >
> > > **About main comment**
> > > Concerning the main comment, I appreciate the reorientation of the paper as well as the changes in wording. You may consider adding a brief overview of how matrix coloring is performed (or at least a good introductory reference) since the motivation of the paper are partially pedagogical.
> > >
> > > **Overall length and content**
> > > I believe the length of at most 12 pages for TMLR mostly ensure fast reviewing (a priori) but I am not sure the final length needs to be 12 pages (the AC may know better). The new experiments are really great and it could be best to include them in the main text as they are making a strong case for sparse automatic differentiation against the most common approach (lazy products).
> > >
> > > **Additional discussuon**
> > > - You could make the case for direct solvers stronger by adding the accuracy of the direct solvers against the accuracy of the iterative solvers in Appendix F 2. The former are generally far superior (as you mention yourself "direct solvers are less vulnerable to ill-conditioning, which occurs frequently in the KKT system of interior point methods")
> > > - For the global sparsity pattern, just to be sure I understood, at tracing, two tracers are propagated, one for the local and one for the global. Each of these tracers follow different rules as explained in table 1.
> > >
> > > Thanks again

---

### Author Response · Authors · 2025-03-28
**Global response**

Dear Action Editor and reviewers,

We are grateful for the detailed feedback, suggestions and criticisms.
We have taken this opportunity to significantly revamp our paper, clarify the contributions and enrich the experimental section.
These improvements are visible on the revised PDF we just submitted.
You will find detailed answers to each review below, but we address common criticisms in the present comment instead.

### Pipeline overview

Some reviewers regretted that our automatic sparse differentiation (ASD) involves two more packages which are not described in the present paper.
We mentioned these connected works for the sake of transparency, but there are several reasons why we believe sparsity detection deserves an independent treatment:

- Sparsity detection is the **main bottleneck** of ASD (see Fig. 2 and 3). Existing techniques in Julia were prohibitively expensive, preventing deployment at scale. Our implementation successfully overcomes this hurdle.
- The literature on AD contains many papers that focus solely on either sparsity detection or coloring. Following this traditional split, there is **only one common author** between the sparsity detection project and the coloring project we mention. The first author of the detection paper has no involvement in coloring and vice-versa, because both projects were developed independently.
- The reason why these projects are independent is that the pipeline we use in our benchmarks is **modular**. It was designed so that various sparsity detection and coloring algorithms could be plugged in seamlessly. This modularity stems from Julia's multiple dispatch paradigm, enabling distributed development instead of monolithic design.
- Finally, sparsity detection is of **independent interest**. As Fig. 4 and 5 showcase, automatically deriving sparsity patterns can illuminate the structure of a model. It can be used to provably detect dependencies between outputs and inputs inside complicated functions without being affected by floating point errors (not unlike interval arithmetic).

As a result, we reoriented the whole paper, including its title, to focus even more on sparsity detection, highlighting our specific algorithmic and software contributions, while purposefully leaving aside the other parts of the pipeline.

### Choice of language and parallelism

Some reviewers wondered why we chose Julia for our method, and why it is limited to the CPU.
We understand that Julia is not the language of choice among many ML practitioners.
There are several reasons why we picked it:

- ASD is very useful in scientific ML, which combines data-driven approaches with physical models (ODEs, optimization, control). These use cases often feature complex, hand-written functions of moderate size, with lots of control flow and scalar indexing. Julia's high performance on non-vectorized code makes it a language of choice for such applications.
- As a result, the Julia community had long had an interest in ASD, whereas it is almost absent from the mainstream Python ecosystem (PyTorch, TensorFlow, JAX), except for the prototypes mentioned in our survey. Still, existing Julia methods had severe drawbacks, which we set out to fix.
- We hope that our implementation can serve as a proof of concept, showing that ASD is possible in Python. In any case, the remaining gap towards a Python implementation seems less daunting than between C++/Fortran and Julia.

As for lacking GPU support, we acknowledge it as a limitation of our approach, and discuss it more in Sec. 4.3.
Porting these methods to GPUs and array-based languages like JAX requires new insights to optimize tensor-level overloads, reason at the block level and avoid dynamically-sized containers.
However, we argue that even a CPU-only library such as SCT can be useful in moderate-sized problems like those we benchmark, especially when sparsity detection can be amortized.
Indeed, in those cases, it does not even need to be run on the GPU.

### Contributions and experiments

Some reviewers asked us to clarify what we bring to the state of the art, in terms of methods and code.
We addressed this by completely rewriting Sec. 4, which now goes into implementation challenges and design choices of our library.
In particular, we insist on the generic way that index sets can be represented, and on the extension to tensor-level overloads in addition to scalar ones.

Finally, some reviewers suggested experiments to demonstrate the effect of ASD on linear solvers, or to highlight more traditional ML applications.
We complied by adding a comparison between iterative and direct solvers on the Brusselator example (appendix F2), and by inserting a new experiment on sparse differentiation of implicit GNN (appendix G).
We also moved up the discussion of ASD's ML applications to Sec. 1.2.
The changes in our revision are highlighted here: https://anonymous.4open.science/r/sparse-differentiation-paper/tracer_paper-diff61c252.pdf

---

### Decision · Action_Editor_1QFb · 2025-04-24

**Recommendation:** Accept with minor revision

**Comment:**

Overall, this paper is of great pedagogical value to the ML community, and while quite narrow, the experiments demonstrate the effectiveness of the proposed package.

The GNN experiment is the one that will speak the most to the ML community, so it should be part of the main text.

I share the reviewer's concern about splitting the contributions: the camera-ready version of the manuscript should include a description of the colouring algorithm; this paper must be self-contained.

**Audience:**

This paper is useful for ML researchers who are working with automatic differentiation, especially regarding its efficiency. Popularizing such ideas might lead to faster and better automatic differentiation algorithms.

**Claims And Evidence:**

This paper is about Automatic Sparse Differentiation (ASD), in which the sparsity of the Jacobian of an operator can be a) detected and b) used to reduce the computational cost of computing the Jacobian by batching orthogonal computations.
The goal of this paper is twofold: it first serves as a useful introduction to the concept of ASD to the ML community, as ASD is seldom used or considered by the ML community. Second, it introduces an operator-overloading-based approach to solve a) efficiently. The efficiency of the proposed approach is demonstrated in computing Jacobians on PDEs, which is vaguely related to ML. Additional experiments on GNN were provided after discussing them with the reviewers, which greatly strengthened the paper.

---

> ### Author Response · Authors · 2025-05-21
> **Camera-ready version**
>
> Dear Action Editor,
>
> Following your instructions, we have prepared the camera-ready version of the paper. It includes the following changes (see the output of `latexdiff` at [this URL](https://anonymous.4open.science/api/repo/sparse-differentiation-paper/file/tracer_paper-diff7e21514.pdf)):
>
> - The manuscript was de-anonymized, authorship contributions and acknowledgements were inserted.
> - The code for the experiments was archived on a [public GitHub repository](https://github.com/adrhill/sparser-better-faster-stronger/) with a [Zenodo DOI](https://zenodo.org/records/15473177), and properly cited.
> - Updated citations were added for the other packages used in the ASD pipeline (the corresponding preprints only appeared this month on Arxiv).
> - GNN experiments were moved to the main text, with additional precisions to facilitate result interpretation.
> - Coloring routines were summarized in a new section in the appendix (Appendix C), with references to the relevant literature for details.
>
> We hope that this minor revision will prove sufficient for the camera-ready version.